# Optical Biosensor Based on Graphene and Its Derivatives for Detecting Biomolecules

**DOI:** 10.3390/ijms231810838

**Published:** 2022-09-16

**Authors:** Guangmin Ji, Jingkun Tian, Fei Xing, Yu Feng

**Affiliations:** School of Physics and Optoelectronic Engineering, Shandong University of Technology, Zibo 255000, China

**Keywords:** graphene, graphene derivatives, optical biosensor, detecting biomolecules

## Abstract

Graphene and its derivatives show great potential for biosensing due to their extraordinary optical, electrical and physical properties. In particular, graphene and its derivatives have excellent optical properties such as broadband and tunable absorption, fluorescence bursts, and strong polarization-related effects. Optical biosensors based on graphene and its derivatives make nondestructive detection of biomolecules possible. The focus of this paper is to review the preparation of graphene and its derivatives, as well as recent advances in optical biosensors based on graphene and its derivatives. The working principle of face plasmon resonance (SPR), surface-enhanced Raman spectroscopy (SERS), fluorescence resonance energy transfer (FRET) and colorimetric sensors are summarized, and the advantages and disadvantages of graphene and its derivatives applicable to various types of sensors are analyzed, and the methods of surface functionalization of graphene and its derivatives are introduced; these optical biosensors can be used for the detection of a range of biomolecules such as single cells, cellular secretions, proteins, nucleic acids, and antigen-antibodies; these new high-performance optical sensors are capable of detecting changes in surface structure and biomolecular interactions with the advantages of ultra-fast detection, high sensitivity, label-free, specific recognition, and the ability to respond in real-time. Problems in the current stage of application are discussed, as well as future prospects for graphene and its biosensors. Achieving the applicability, reusability and low cost of novel optical biosensors for a variety of complex environments and achieving scale-up production, which still faces serious challenges.

## 1. Introduction

Since the isolation of graphene in 2004, two-dimensional (2D) layered materials have attracted great attention [1]. Compared with traditional bulk materials, 2D nanomaterials and their nanocomposites exhibit excellent physical, chemical, optical and electronic properties, which are widely used in reaction catalysis, energy storage, biosensing, drug transport and biological therapy [2,3,4]. In this context, graphene and graphene derivatives, such as graphene oxide (GO), reduced graphene oxide (rGO), and graphene quantum dots (GQDs), have many excellent properties, which make them widely used in biosensing applications. Such as a large specific surface area, which provides an extremely high density of surface-active sites; a surface containing oxygen functional groups makes ease of chemical modification; and efficient energy transfer, and biocompatibility. In the face of the rampant new coronavirus, the world is looking for a rapid detection method, and the graphene-based biosensor excels in the rapid detection of COVID-19 virus, which is not only fast and reliable, but also convenient and fast [5]. The results of graphene-based ultra-surface microfluidic biosensors for antibiotics and DNA show the superiority of the sensors for terahertz biosensing, opening up exciting prospects for terahertz label-free biosensing [6]. In particular, graphene and its derivatives possess unique optical properties, such as broadband and tunable absorption, fluorescence quench, and strong polarization-dependent effects; this excellent optical property lays the foundation for the establishment of optical biosensors based on graphene and its derivatives.

Optical biosensors offer significant advantages over traditional analytical sensing technologies, enabling direct, real-time and label-free detection of biological and chemical substances. Along with the advances in emerging science and technology, a variety of cross-disciplinary and advanced approaches (e.g., microelectronics, nanotechnology, molecular biology, etc.) are applied to novel optical biosensors. Optical detection is achieved by using the interaction between the optical field and the biometric element. There are two main modes of optical biosensing: label-free mode and label-based mode. In the label-free mode, biosensing is achieved based on signals generated directly from the interaction of the analytic material with the transducer. Label-based mode sensing achieves biosensing by using a label and then generating an optical signal using methods such as colorimetry, fluorimetry, or luminescence. Biomolecular detection is widely used in food safety, environmental monitoring, disease prevention and clinical treatment [7]. In particular, biosensors based on 2D materials have been widely used to detect biomolecules, such as nucleic acids [8], viruses [9], uric acid [10], bacteria [11], proteins [12], glucose [13] dopamine [14]. Meanwhile, optical biosensors have the advantages of high sensitivity, good specificity, small size, cost-effectiveness, and no damage to the detected cells. For example, handheld glucose meters used by diabetic patients are the most successful commercial application of biosensors, where glucose can be detected oxidatively by labeling-assisted sensing enzymes with fast detection, high sensitivity, small size, and simplicity of operation.

In the past decade, biosensing applications based on two-dimensional materials have developed rapidly [15,16]. In this review, we discuss the synthesis and engineering methods of graphene and its derivatives for biosensing applications. Then, we focus on graphene and its derivatives optical sensors, FERT biosensors, SPR biosensors, SERS sensors, colorimetric biosensors, and the interaction between graphene and its derivatives and biological cells. Finally, we look forward to the future of graphene and its derivatives in biosensing and biomedical fields.

## 2. Synthesis and Preparation of Graphene and Its Derivatives

The preparation method of 2D materials plays an important role in whether they are suitable for biosensors, biomolecular detection and biomedical therapy. Graphene and its derivatives mainly include four forms, which are graphene itself, GO, rGO and GQDs. In this section, we focus on the synthesis of these 2D nanomaterials.

Graphene is a two-dimensional single-atom layer material consisting of six bonded sp^2^ hybridized carbon atoms [17,18]. Among all sp^2^ carbon isomers, graphene exhibits the most remarkable properties; it has a remarkable theoretical specific surface area and a very high Young’s modulus [19], ultra-high electron mobility [20], extremely high thermal conductivity [21] and electrical conductivity [22]. Due to the presence of benzene rings, graphene readily forms covalent or non-covalent bonds with other atoms, altering its electrical or chemical properties [23]. Currently, the common methods for preparing graphene include mechanical exfoliation, chemical vapor deposition (CVD), epitaxial growth, etc.

GO is an oxidized form of graphene, which contains oxygen functional groups on its surface [24,25]. GO has both the hydrophobicity of pristine graphite and the hydrophilicity of oxygen functional groups [26,27]. In addition to the excellent properties of graphene, GO has properties such as fluorescence quench, efficient energy transfer, biocompatibility, and ease of chemical modification. Combining these properties with a simple and scalable preparation process makes GO potentially applicable in many fields [28]. The main methods for the preparation of graphene oxide include Hummer’s method, modified Hummer’s method, microwave-assisted method, and mild thermal annealing method and so on.

RGO is a chemical derivative of graphene after redox treatment. RGO has graphene-like properties, including relatively good electrical conductivity and exhibits good absorption properties throughout the spectrum [29,30,31,32]. Because its constituent graphene layers are functionalized with epoxy and hydroxyl groups, it can be dispersed in a variety of solvents. Reduced graphene oxide is simple to prepare and can be easily prepared from inexpensive graphene oxide by using various (electro)chemical, microwave, and light-assisted thermal methods [33,34,35,36]. In addition to the preparation of rGO using chemical reagents, thermal reduction, low-temperature heat treatment, SO_2_ reduction, UV radiation, and microwave assistance are also common methods for the preparation of rGO.

GQDs can be described as sub-100 nm nanoparticles consisting of a single layer or several layers of graphene, which display quantum confinement in all three spatial directions. The desired properties can be obtained by altering the electronic properties of GQDs by adding dopants to change the chemical composition of GQDs and the edge passivation of various functional groups [37,38]. The tunable optical properties of GQDs can be altered by tuning the physical and chemical properties. As the diameter of GQDs increases and the surface is passivated by functional groups, the emission spectrum tends to be red-shifted, and the fluorescence is enhanced [39,40,41]. GQDs contain hydrophilic functional groups at the edges, which have excellent water solubility and allow the use of materials to modify them, showing edge effect characteristics. GQDs consist entirely of carbon elements, exhibit good solubility in water and become a non-toxic alternative to metal-based nanoparticles with good biocompatibility [42,43]. At present, GQDs preparation methods mainly include top-down methods and bottom-up methods. The top-down methods mainly include hydrothermal/solvothermal, liquid phase exfoliation and electrochemical exfoliation of graphene. The bottom-up methods mainly include solution chemical synthesis, microwave synthesis and cage opening of fullerenes.

### 2.1. Synthesis of Graphene

#### 2.1.1. Mechanical Exfoliation Methods

In 2004, Scientists in the UK successfully prepared graphene by mechanical stripping, using adhesive tape repeatedly bonded from the lump of graphite to produce graphene [1]; this mechanical stripping method can be used to produce high-quality 2D nanofilms with very low cost and relatively simple operation [44]. Yang et al. developed a new method to prepare small amounts of graphene by controlling the homogeneity of graphene through a “tape peeling” method (Figure 1a) [45]; this method improves the uniformity of graphene and helps to control the thickness of graphene by removing a large amount of residue from the direct graphene growth process by exfoliation. Recently, taking into account the interactions between graphene molecules, a method has been proposed to strip graphene from polymer tape, as shown in the diagram below (Figure 1b) [46]. Graphite containing multiple graphene sheets is compressed between two polymer layers under isothermal and constant pressure conditions. By increasing the height between the polymer layers at a constant rate, the graphite is exfoliated to obtain high-quality graphene in the gauge set. Although this method is generally widely used to prepare high-quality graphene samples, it has the disadvantages of being time-consuming and long, high labor intensity and low yield. However, graphene materials prepared by mechanical stripping have no choice of size and thickness uniformity, and graphene film generation is very random. Graphene produced by mechanical stripping is therefore used to study basic properties and proof-of-concept applications; often prepared in the laboratory, it is not suitable for mass production [47,48,49].

#### 2.1.2. Chemical Vapor Deposition (CVD) Methods

In chemical vapor deposition (CVD), under very high temperatures and specific gas environment conditions, the reaction substance has a chemical reaction, the decomposition of the substance deposited on the substrate surface, to get different thickness but good uniformity of solid material film [50]. Zhao et al. achieved the first self-limited growth behavior of monolayer graphene on electroplated copper foil prepared with carbon precursors other than methane using low-pressure chemical vapor deposition (LPCVD) technology (Figure 2a) [51]. A high degree of similarity was found between graphene prepared using ethanol and graphene grown with methane, suggesting that the self-limiting growth on copper surfaces using LPCVD is independent of the precursor structures of ethanol and methane when the carbon flux is sufficiently low. By passing carbon-containing gas through a catalytic converter, a large area, of high-quality graphene films is produced on the surface of the substrate. The preparation of graphene on crystal surfaces of different metals and metal carbides by CVD requires a reaction chamber with a high vacuum. Graphene grown by CVD can be divided into two types in principle, namely, dialysis mechanism and surface catalytic mechanism. The difference between the two mechanisms is the amount of carbon in the metal substrate. Li’s group studied the growth of graphene on Cu, Ni and Cu/Ni surfaces by using carbon isotope labeling technique (Figure 2b) [52]. Due to the catalytic activity of the metal surface and the high solubility of carbon, the decomposition of methane provides ^12^CH_4_ or ^13^CH_4_, which rapidly diffuses into the metal. When the carbon in the metal reaches supersaturation at a certain temperature, the graphene film forms on the surface of the metal substrate due to the segregation of the carbon atoms. The solubility of carbon in different metals is very different. For example, the solubility of carbon in nickel is much higher than that of copper. At 1000 °C, the solubility of carbon in nickel and copper is 1.3 at % and less than 0.001 at %, respectively [53,54]. In addition, the carbon concentration and cooling rate directly determine the number of graphene layers deposited on the metal surface. The morphology of the metal substrate is also a key parameter in the preparation of graphene films. Huet et al. showed that graphene grown on epitaxial copper film had superior flatness, more consistent contact with the final substrate, and stronger adhesion. In contrast, polycrystalline copper templates cause graphene to exhibit significant out-of-plane deformation, prone to cracking, wrinkling, uneven strain and doping changes [55]. In addition to metal substrates, it is also possible to synthesize graphene on insulating substrates. Tai et al. realized the epitaxial growth of graphene inverted on a monocrystalline silicon substrate at a metal-free atmospheric CVD temperature of 900–930 °C (Figure 2c) [56]. Due to the catalytic sluggishness of silicon, the thermal decomposition of methane produces active carbon atoms, which trigger the nucleation of graphene. When the substrate is inverted, the high saturation and collision frequency of carbon radicals can enhance the nucleation of graphene. At higher temperatures, as the silicon surface becomes active, graphene begins to spread and grow around its edges, forming concave, double-layer regions of larger size. In short, layers of graphene can be deposited on an insulating substrate without catalysis. To produce high-quality graphene, higher decomposition temperatures are required on insulating substrates than on metallic substrates such as Cu or Ni [57]. Graphene grown by CVD has not only high electron mobility and conductivity, but also excellent transparency and uniformity.

#### 2.1.3. Epitaxial Growth Methods

The basic mechanism for epitaxial graphene growth on SiC is simply heating the substrate (in a vacuum or inert atmosphere) to a temperature normally in the range of 1200 °C to 1800 °C. At these temperatures, the surface layer of the SiC crystal is thermally decomposed, causing the silicon atoms to sublimate. The remaining carbon atoms on the surface of the SiC crystal rearrange, recombine, regraphitize and form epitaxial graphene layers [58]. The growth of epitaxial graphene on the Si-face SiC surface is different from that on the C-face SiC surface. Monolayers of epitaxial graphene on both surfaces show the expected electronic structure and transport properties of graphene, but the non-graphite stacking of multilayers on C-face SiC determines that the electronic structure is quite different from that of graphite multilayers on Si-face SiC. For example, graphene grown on the Si-face exhibits charge-carrier mobility of 500–2000 cm^2^ V^−1^ S^−1^, while graphene grown on the C-face can reach 10,000–30,000 cm^2^ V^−1^ S^−1^. Yang et al. successfully epitaxial growth of large-area graphene single crystals with fixed stacking orientation on h-BN using the plasma-assisted deposition method for the first time [59]. During growth, the feedstock gas methane (CH_4_) is decomposed into various reactive free radicals, allowing graphene to nucleate and subsequently grow at its edges [60,61]. Hydrogen atomic is also produced in the CH_4_ plasma, which is an important etcher for removing amorphous carbon and ensuring the growth of sp^2^ carbon. Zhang et al. reported a novel method for the preparation of ultra-flat single crystal graphene using Cu/Ni (111)/sapphire wafers at lower temperatures, reducing the temperature of epitaxial grown graphene to 750 °C, much lower than the catalytic surface temperature reported earlier (Figure 3a) [62]. The quality of graphene is as good as that of graphene grown at high growth temperatures, with carrier mobility up to ≈9700 cm^2^ V^−1^ S^−1^ at room temperature, opening up a new method for high-quality graphene growth. Jin et al. presented a practical and effective method for the growth of uniform graphene by 4H-SiC (0001) thermal decomposition epitaxy assisted by nitrogen plasma (Figure 3b) [63]. With the help of nitrogen plasma, the number of graphene layers can be precisely controlled, and high-quality monolayer and bilayer graphene with a large-scale uniform surface has been successfully obtained. With proper modification, single-crystal Cu (111) foils and single-crystal graphene films can be generated, thus reaching a very large scale, allowing industrial and large-scale production of large-area high-quality graphene films with single-crystal properties (Figure 3c) [64].

### 2.2. Synthesis of GO and rGO

#### 2.2.1. Hummers and the Modified Hummers Method

The most important branches of graphene derivatives are GO and rGO. Graphite goes through a series of oxidation reactions to form GO, and reduction of GO produces rGO; thus far, GO has been prepared mainly by Boride [65], Staudenmaier [66], and Hummers methods [67], and the improved scheme based on these three methods. The above preparation methods all involve the oxidation of graphite to different degrees. Boride reacted with graphite in a mixture of nitric acid (HNO_3_) and potassium chlorate (KClO_3_) to form graphite oxide [65]. Later, Staudenmaier improved on Boride’s method by adding chlorates to the reaction process. Sulfuric acid is also added to the reaction medium to increase its acidity of the reaction medium. Staudenmaier’s method is similar to Boride’s method in that a slight modification in the reaction process leads to the overall oxidation of graphite, making the oxidation reaction more efficient in a single container [66]. In 1958, Hummers developed another new method of synthesizing GO using a mixture of sodium nitrate (NaNO_3_), concentrated sulfuric acid (Con H_2_SO_4_) and potassium permanganate (KMnO_4_) as an oxidant, which is the most widely used GO synthesis method [67]. The graphite salt produced in this method acts as a precursor of GO and is then stripped in a solvent by ultrasound. In addition to the more familiar methods mentioned above, the researchers have also developed some improvements based on the above methods, such as the addition of additional oxidants in the synthesis step. The end product of the reaction depends not only on the specific oxidizer used, but also on the reaction conditions.

Different preparation methods have their own advantages and disadvantages. Boride’s and Staudenmaier’s methods use nitric acid and KClO_3_ to produce harmful gases, and Staudenmaier’s method has a long reaction time [66,68]. Hummer’s method avoids these problems and is widely used. However, this method can cause heavy metal pollution due to the addition of heavy metal elements. In addition, the products can contain ions that are difficult to remove. Besides, the oxidant in concentrated sulfuric acid has strong oxidation performance, potassium permanganate and concentrated sulfuric acid reaction product Mn_2_O_7_ in contact with organic compounds or with a heating temperature of higher than 55 °C will explode [69,70]. Subsequently, Marcano and his team modified Hummer’s method to synthesize GO (Figure 4a) [71]. The process removes NaNO_3_ from the mixture and replaces it with less corrosive phosphoric acid (H_3_PO_4_) to produce GO, which does not involve the release of strong exothermic or toxic gases. By increasing the content of KMnO_4_ and adjusting the ratio of Con H_2_SO_4_ to H_3_PO_4_ in the mixture, the oxidation efficiency can be significantly improved, and the prepared GO has a high oxidation degree and regular structure.

#### 2.2.2. Other Methods of GO Preparation

In addition to Hummers and the modified Hummers method, the researchers have also proposed microwave-assisted preparation of GO. For example, Wang et al. successfully prepared nanosheets to go using microwave radiation. The method first mixed H_2_SO_4_, graphite sheet and KMnO_4_ in a three-neck flask and placed in a microwave oven for 150 s. Then, hydrogen peroxide was added and placed in a centrifuge for centrifugation. Finally, the supernatant was removed, and the underlying mixture was ultrasonically treated to obtain a uniform suspension of GO [73]. In addition to the microwave synthesis method, Grossman and his colleagues proposed a simple mild thermal annealing method to synthesize GO [74]; this method controls the annealing temperature at 50–80 °C, does not involve chemical treatment, and avoids heavy metal pollution and harmful gas generation. Thermal annealing can promote the transformation of the sp^2^–sp^3^ hybrid phase into an obviously oxidized ordered graphite phase. Thermal annealing can control the oxygen functionalization of GO to produce go with enhanced electronic properties [75]. In particular, the chemical oxidation of graphite requires small particles of the raw material, which often requires intense agitation or even ultrasonic treatment in order for the chemical reaction to be sufficient; these factors make it difficult to obtain large flakes of GO. Dong et al. proposed an improved go preparation method to prepare large-sized and super-sized GO flakes [76]. The preparation of GO is carried out in two steps. The first step is to chemically peel the raw graphite into graphene sheets in a liquid medium. The second step is to oxidize the stripped graphene sheets under mild conditions. The core of this preparation method is the chemical stripping of raw graphite into graphene sheets. For example, the interaction of chromium trioxide, hydrogen peroxide and crystalline graphite can be stripped at room temperature to produce uniform graphene sheets [77]. The change of preparation conditions ensured the integrity of the prepared graphene flakes. Compared to conventional Hummer, KMnO_4_ is used in a lower concentration. The peeled graphene sheets were oxidized to obtain a large area of uniform GO sheets, avoiding the formation of oxidation defects (Figure 4b) [71,72,78].

#### 2.2.3. Methods of Preparing rGO

So far, there are few reports on the synthesis methods of rGO. Most researchers use hydrazine hydrate to reduce graphene oxide [79,80,81]. However, due to the high toxicity of the reductant, this method cannot be used to prepare rGO on a large scale. In addition, some other reducing agents are also used to synthesize rGO. Paredes et al. compared the deoxygenation efficiency of different reducing agents for GO suspensions and the deoxygenation efficiency of suspensions under alkaline heating conditions and found that only vitamin C was competitive with widely used but highly toxic hydrazine in reducibility (Figure 5a) [82]; this method extends the preparation conditions from water to some common organic solvents to prepare stable rGO suspensions. Because vitamin C only contains C, H and O elements, the use of vitamin C as a reducing agent avoids the introduction of other elements into the reduction product and improves the purity of the product. Su et al. obtained highly conductive reduced GO by reducing it with ethanol vapor at high temperatures [83]. Reduction of GO with ethanol is a relatively mild reduction method that does not cause serious damage to the edges of GO and ensures the relative integrity of the reaction products. In addition, the reduction of GO by H_2_ was also studied under the same conditions. The results show that at the same temperature, ethanol vapor can reduce GO more effectively. In addition to using alcohol as a reducing agent, the researchers also used phenols to reduce GO. Wang et al. investigated the reduction effect of GO-containing hydroquinone [84]. The reaction produces rGO with an orderly crystal structure and low oxygen content, but due to its structural characteristics, it is easy to agglomerate in water, and the thermal stability of the product is poor compared with the original graphite.

In addition to the use of chemical reagents to reduce GO, the thermal reduction is also a common method for the synthesis of rGO. Thermally reduced graphene oxide (TRG) is prepared by placing dry GO in a high-temperature inert gas environment and heating it rapidly. Heating rate, reduction temperature and heating time directly affect the properties of thermal reduction products. The principle of thermal reduction is to rapidly anneal the reactants at high temperatures, and to achieve the separation of the target products by decomposition of oxygen functional groups to produce CO or CO_2_ gas expansion. In order to obtain rGO with good performance, it is recommended to control the reduction temperature above 1000 °C. Hyunwoo Kim et al. showed that GO was placed in inert gas and heated to 1000 °C for 30 s to synthesize TRG [86]. Stripping occurs when the pressure of the gas (CO_2_) produced by the decomposition of the epoxy and hydroxyl sites of GO exceeds the van der Waals forces that pull the GO sheets between them. Fadil et al. proposed a multifunctional and environmentally friendly method for the synthesis of rGO [87]. In water-based fine emulsions, GO nanosheets and traditional surfactants sodium dodecyl sulfate (SDS) are free radical polymerized to form polymer/GO films at ambient temperatures without the use of special techniques such as hot pressing. Subsequently, the film was heat treated at 160 °C to reduce GO to rGO. In addition to the above studies, other factors affecting the products of thermal reduction reactions have also been investigated by some researchers. Kumar et al. investigated the effect of oxygen clusters on reduction products during thermal reduction [88]. The experimental results show that the clustering degree of oxygen affects the ratio of oxygen and carbon atoms in the rGO reduction process but does not affect the reduction temperature. Shen et al. and Niu et al. studied the effect of low-temperature heat treatment on the preparation of rGO (Figure 5b) [85,89]. Niu et al. successfully synthesized rGO sheets by low-temperature heat treatment of GO sheets at 200 °C for 2 h [85]. Shen et al. demonstrated that cryogenic reduction of GO involves the following four steps. Step 1: below 160 °C release of physical adsorption water; Step 2: at 160–210 °C, the functional groups decrease; Step 3: at 210–300 °C, sulfate, stable oxygen and aromatic by-products are generated; Step 4: above 300 °C, the carbon network and oxygen function degrade. The formation of gaseous aromatic byproducts below 300 °C indicates that the GO carbon plane is decomposed even at low temperatures (below 300 °C); this is in contrast to previous experiments that showed that the GO carbon plane decomposed only at higher temperatures (above 350 °C) [89].

Besides, the well-known synthesis methods mentioned above, there are other reduction methods, such as SO_2_ reduction, ultraviolet radiation reduction, and microwave reduction. Researchers found that sulfur-containing compounds had a good reducing performance. The oxygen content of the reduction product obtained by using SO_2_ gas to reduce GO was similar to that obtained by using hydrazine as a reducing agent, and the use of SO_2_ promoted the dispersion of rGO in an aqueous solution. The ultraviolet radiation reduction method has great potential in the large-scale synthesis of rGO. Williams et al. irradiated GO suspension with ultraviolet light to obtain stable reduction products [90]. Park et al. successfully obtained rGO thin films at room temperature and environmental conditions by shining flashlights on GO thin sheets [91]. Compared to traditional methods of reducing graphene oxide using high temperature or strong reducing agents, this soft reduction method is not only simple and inexpensive but can be carried out at room temperature without the use of chemical agents. Xie et al. reviewed the microwave-assisted preparation of reduced go, which can be divided into three categories: (1) microwave-assisted chemical reduction; (2) microwave-assisted thermal reduction; (3) microwave-assisted simultaneous thermal stripping and thermal reduction; these methods open up new avenues for GO reduction [92].

### 2.3. Synthesis of Graphene Quantum Dots (GQDs)

#### 2.3.1. Top-Down Methods

The top-down method mainly decomposes bulk materials by physical or chemical methods to prepare GQDs. The first GQD was discovered in this way, the top-down approach plays an important role in discovering new materials and studying their structural and properties. The properties of GQDs can be controlled by changing the synthesis method and reaction conditions. Among the top-down methods, hydrothermal, electrochemical, liquid exfoliation and photolithography are commonly used.

##### Hydrothermal/Solvothermal Methods

Hydrothermal/solvothermal synthesis is one of the most common ways to synthesize GQDs [93]. Wu et al. synthesized GQDs from GO by hydrothermal method (Figure 6a) [94]. Large sheets of GO are cut into small flakes by controlled oxidation in a mixture of H_2_SO_4_ and HNO_3_ under mild ultrasound. Then it was placed at 200 °C for 10 h to obtain GQDs. Shen et al. passivated the surface of GQDs by hydrothermal reaction using small GO slices and polyethylene glycol (PEG) as raw materials [95]. The key to the method is to treat the GO flakes with PEG and then reduce them with hydrazine hydrate. Then, a more efficient hydrothermal method was developed to reduce GO flakes to GQDs. The proposed principle for the synthesis of GO into GQDs is the presence of oxygen-containing epoxy and carboxyl groups in linear chains within the lattice, which is an unstable structure. To form a more stable state, the underlying C-C bonds in the graphene lattice break, and the lattice is cut along this chain [94,96]. To investigate the effects of precursors and temperature on GQDs synthesis, Xie et al. treated three different precursors (CNTs, GO and carbon black) with hydrothermal treatment in a mixture of ethanol and hydrogen peroxide. By Fourier transform infrared spectroscopy and Raman spectroscopy measurement, GQDs obtained by the three have obvious defect structure, PL measurement results also have obvious differences, which proves that the PL of GQDs is closely related to its defect structure (Figure 6b) [97]. Hydrothermal/solvothermal synthesis of GQDs is facilitated by providing energy at high temperatures and dispersing the medium. The researchers studied the oxidation effect of the precursor and the properties of the product during synthesis. Before hydrothermal treatment, GO goes through reflux, HNO_3_ oxidation and dialysis to obtain primary GQDs (GQD1). The collected primary GQDs were refluxed again in HNO_3_, and the obtained solution was dialyzed to obtain the second batch of GQDs (GQD2). The GQDs obtained by secondary reflux is smaller than the former in terms of material size and can show the photoluminescence (PL) effect at longer wavelengths (Figure 6c) [98]. Recent research in hydrothermal synthesis has focused on improving the ease of synthesis and avoiding environmental pollution from the use of strong oxidants. Kellici et al. proposed a fast and environmentally friendly method to synthesize GQDs by combining hydrothermal synthesis with flow synthesis [99]. Fluid properties synthesis conditions allow for simple modifications. Due to the presence of oxygen-containing functional groups at the edge of the raw material, it is highly water-soluble, allowing the large-scale synthesis of GQDs. However, the hydrothermal/solvothermal synthesis method requires higher reaction conditions and is easy to cause environmental pollution problems. In addition, hydrothermal/solvothermal synthesis cannot control the size, morphology and properties of the reaction products due to the random cleavage of chemical bonds during the reaction. Recently, researchers have been trying to overcome these problems by changing the reaction conditions to achieve a directly controlled synthesis of GQDs.

##### Electrochemical Exfoliation Methods

Electrochemical exfoliation, a method of obtaining layered bulk materials with the help of an applied electric field, was first used to obtain raw graphene sheets because of their ability to control product size and shape, and more recently for the synthesis of GQDs [100,101]. The electrochemical stripping mainly consists of two steps: first, an anodic potential is applied to the reference electrode to cause the chemical bond of the bulk material to break and the electrolyte to intercalate between the bulk material layers. Electrochemical reduction then occurs at the cathode (Figure 7a) [102]. Compared to hydrothermal/solvothermal synthesis, electrochemical stripping allows selective oxidation controlled oxidation and does not require toxic chemicals and can be performed under environmental conditions. Pang et al. electrochemically oxidized graphite to GQDs using platinum wire counter electrode in phosphate solution [103]. Chi’s team electrochemically synthesized GQDs in neutral phosphate solution in an electrochemical cell consisting of graphite rod, platinum net, and silver/silver chloride as working electrode, a counter electrode and reference electrode, respectively [104]. Li et al. treated graphene films with plasma to increase hydrophilicity. The electrochemically synthesized GQDs had high stability in a water medium and could be preserved for a long time [105]. Huang et al. developed an electrochemical method based on a weak electrolyte solution to obtain high-yield GQDs. The GQDs yield of this method is tens of times higher than that of GQDs prepared by strong electrolyte, with short preparation time and high crystallinity of the synthesized product (Figure 7b) [106]. Kumar et al. synthesized GQDs from waste batteries using citric acid and NaOH as electrolytes and graphite rods from waste batteries as electrodes. By applying a voltage, the color of the electrolyte solution changed to dark brown, demonstrating that the graphite rod was shed, followed by dialysis, centrifugation and washing to obtain GQDs (Figure 7c) [107]. Joffrion et al. synthesized GQDs independently excited by microplasmas-assisted electrochemical synthesis. The method can achieve excitation independence through a high level of oxygen passivation, and can emit shorter wavelengths by simple dilution, with the ability to emit any desired color [108]. The change of electrolyte solution further realizes the functionalization of GQDs synthesized by electrochemical synthesis. The relatively simple reaction device, no strong oxidant and more accurate synthesis of GQDs make electrochemical synthesis an effective synthesis process for GQDs production.

##### Liquid Phase Exfoliation Methods

Another method to synthesize GQDs is liquid phase exfoliation (LPE). The method uses ultrasound to decompose macromolecular materials into nanoscale particles without the need for oxidation reagents, which can effectively maintain the integrity of the carbon source and has strong scalability. Lu et al. used acetylene black and nano graphite as raw materials, dissolved in organic solvent N-methyl-2-Pyrrolidone, and treated by ultrasound for one hour, respectively, to obtain high defect GQDs and low defect GQDs. The precursor material structure results in significant differences in the size and density of the synthesized GQDs defects (Figure 8a) [109]. Hubalek combines microwave expansion with low-energy LPE to produce blue luminescent GQDs. Microwave treatment causes the graphite to expand, and LPE separates the graphite by ultrasound. The prolongation of the acoustic wave action causes the shedding of the ultra-small graphene flakes and the formation of GQDs in various surface states within the flakes (Figure 8b) [110]. The principle of LPE is that solvent molecules between graphite sheets vibrate under ultrasonic action, leading to structural defects and further decomposition of graphite sheets. The microwave produces the vibrations needed for LPE, and the microwave radiation causes the graphite sheets to expand, facilitating the intercalation of solvent molecules between the graphite sheets. Compared with the electrochemical synthesis method, because the intercalation positions of solvent molecules in the lattice are randomly dispersed, the stripping of large pieces of material is also random, and LPE cannot control the state of GQDs generation. However, LPE is safer, simpler, and more environmentally friendly than other top-down methods and can be used for the synthesis of scalable, high-yield GQDs.

#### 2.3.2. Bottom-Up Methods

The bottom-up approach is to assemble GQDs from molecular precursors, and the diversity of carbon precursors determines the size and performance of the synthesized GQDs. Among the bottom-up methods, cage opening of fullerenes methods, stepwise organic methods and microwave-assisted methods are commonly used.

##### Cage Opening of Fullerenes Methods

Loh et al. reported an open-cage synthesis of GQDs by metal-catalyzed fullerenes; they used fullerene as the reactant, and through the strong interaction between carbon atoms of fullerene on the ruthenium surface, vacancies appeared on ruthenium surface, which contributed to the embedment of fullerene molecules, carbon clusters occurred at high temperature, and GQDs with different shapes were formed through diffusion and aggregation [111]. Chua et al. treated Buckminster Fullerene with strong acid and chemical oxidant, induced its oxidation, cage opening and fragmentation, and prepared ultra-small GQDs [112].

##### Stepwise Organic Methods

Stepwise organic synthesis is the synthesis of GQDs from small carbon precursors by organic reactions, and the structure of the synthesized product can be controlled. Yan et al. prepared GQDs by stepwise organic synthesis, and their quantum dots were composed of 168, 132 and 137 conjugated carbons formed by the oxidation of polystyrene dendritic precursors. The synthesized GQDs are very stable, and the method can control the properties of the synthesized GQDs according to the requirements (Figure 9a) [113,114]. Tian et al. reported another stepwise organic synthesis method, in which nitrogen-doped GQDs were formed by dehydration polymerization of organic nitrite acetic acid under high temperature and pressure [115]. The method works by forming double bonds between carbon atoms through nucleophilic addition after the dehydration step. The GQDs synthesized by this method are larger than the GQDs previously reported. Due to nitrogen doping, GQDs show blue fluorescence and have a larger band gap.

##### Microwave-Assisted Methods

Hydrothermal synthesis of GQDs is time-consuming and not suitable for large-scale industrial production. The method of microwave-assisted synthesis of GQDs can accelerate the synthesis speed, heat evenly, and the size distribution of precursor is uniform. The main advantage is that GQDs can be synthesized without any passivating agent. Using aspartic acid and ammonium bicarbonate as raw materials, Zhang et al. prepared blue luminescent GQDs by microwave-assisted synthesis at 560 W working power (Figure 9b) [116]. Zhao et al. reported green fluorescent GQDs extracted from deoiled asphalt using a one-step microwave-assisted method at 600 W power [117]. Kumawat et al. reported a pot of microwave-assisted green synthesis route for the synthesis of red glowing GQDs from ethanol extract of mango leaves [118]. Microwave-assisted hydrothermal synthesis is a simple and feasible method for GQDs synthesis, which is an improvement on the traditional heating method. Microwave-assisted hydrothermal synthesis can be heated synchronously, uniformly and rapidly, resulting in uniform size distribution and non-surface passivation of quantum dots.

In this section, we summarize and discuss the common methods for the preparation of graphene and its derivatives and analyze the advantages and disadvantages of each method. We can decide the method to prepare the samples based on the desired target properties and the advantages and disadvantages of various preparation methods, and present them in Table 1.

## 3. Bio-Functionalization of Graphene and Its Derivative

The methods for functionalizing graphene-based nanomaterials with biomolecules can be divided into two categories, non-covalent modification and covalent functionalization, according to the principle of interaction. Both methods have their advantages and disadvantages. For example, non-covalent crosslinking does not affect the intrinsic properties and structure of nanomaterials, nor does it affect the activity of biomolecules [119]. The preparation process is simple and convenient, but the stability of the product is poor, and false-positive signals are likely to occur. The covalent method uses chemical methods to stabilize biomolecules on the surface of graphene and its derivative materials, which makes up for the lack of non-covalent methods. However, more in-depth studies are needed to obtain higher coupling efficiency and to reduce the damage to the electronic structure and function of graphene and its derivatives. In order to achieve different sensing effects, we can choose different preparation methods in the process of functionalization of graphene and its derivatives.

### 3.1. Non-Covalent Methods

Usually, the simplest and fastest way to prepare biomolecule-functionalized graphene and its derivative complexes is the direct physical adsorption of biomolecules on the surface of graphene and its derivative materials without the aid of external conditions. The electrostatic, π-π stacking, hydrogen bonding and hydrophobic forces between biomolecules and graphene and its derivative materials play an important role in facilitating their interactions. Due to the large two-dimensional aromatic surface, graphene materials can be tightly bound to molecules containing aromatic rings by π-π superposition [120]. Therefore, most biomolecules (e.g., DNA, antibodies, etc.) can be directly adsorbed on the graphene surface [121]. The adsorption of biomolecules by graphene and its derivative materials can also be achieved using electrostatic interactions [122,123]. GO and rGO are negatively charged due to their oxygen-containing groups, so they can adsorb positively charged biomolecules [124]. For negatively charged biomolecules, adsorption can be achieved by functionalizing polymers such as polyethyleneimine and polyaniline to positively charge-neutral graphene, or even negative GO/rGO [125]. In addition, both pristine graphene and GO contain hydrophobic domains, and thus they can be non-covalently bound to certain biomolecules containing hydrophobic groups through hydrophobic interactions with each other [126]. Hydrogen bonding is the interaction between electronegative atoms and hydrogen atoms, which was found to exist at the interface between biomolecules and graphene materials [127]. Typically, the synergistic involvement of two or more forces in the non-covalent self-assembly of biomolecular-functionalized graphene. Alternatively, metal nanoparticles (e.g., AuNPs, AgNPs, PtNPs) can also be used as modifiers of graphene-based nanomaterials (especially GO and rGO) linkers to achieve non-covalent functionalization [128]; These metal nanoparticles can not only achieve stable connections between graphene and biomolecules, but also enhance and amplify sensing signals.

### 3.2. Covalent Methods

Given the high stability of covalent bonding, covalent modification methods are of great importance for functionalized graphene nanomaterials of biomolecules. Since the surfaces of GO and rGO contain oxygen-containing functional groups such as carboxyl and hydroxyl groups, they can effectively immobilize biomolecules containing a variety of functional groups [129]. Most biomolecules (e.g., proteins, enzymes, antibodies, peptides, etc.) contain a large number of amino groups, which form stable amide bonds with the assistance of carboxyl groups on the surface of graphene oxide or reduced graphene oxide, leading to biomolecular-functionalized nanomaterials [130]. For some biomolecules without amino functional groups, such as nucleic acids, similar coupling effects can be achieved by tagging the amino group at the end. In addition, polyaniline not only enhances the physical adsorption of biomolecules with graphene and its derivatives to achieve non-covalent functionalization, but also provides amino groups to provide the conditions for covalent coupling [131]. Some bifunctional molecules, such as glutaraldehyde, 1-pyrenebutyric acid N-hydroxysuccinimide ester (PNHS) and 1-pyrenebutanoic acid succinimidyl ester (PASE), can couple graphene and its derivative materials to biomolecules via bifunctional groups for effective covalent functionalization [132].

## 4. Optical Biosensor

Optical biosensors have high sensitivity and can even detect single molecules. Currently commonly used optical biosensors are mainly based on the following methods, surface plasmon resonance (SPR), surface-enhanced Raman spectroscopy (SERS), fluorescence resonance energy transfer (FRET) and colorimetric (Figure 10) [133]. SPR sensor is realized by the local refractive index change caused by the target adsorbed to the sensor surface [134]. The adsorption of target molecules on the surface of the sensor will cause the enhancement of Raman scattering, which is the principle of the SERS sensor [135]. The signal source of SRES sensing is the inelastic scattering of photons (typically from lasers) when they are incident with molecules [136]. FRET is a dipole coupling between two fluorophores to form an energy transfer mechanism to achieve sensing [137]. Fluorescent sensors are sensitive to changes in the distance between fluorophores. Therefore, changes in fluorescence intensity due to the target analyte have become a commonly used method to detect the target analyte. Colorimetric biosensors enable detection by using enzymes or enzyme-like materials to catalyze the chemical reactions that accompany color changes. In this section, research reports on optical biosensors for in vitro detection of proteins, glucose, metal ions, hormones, DNA, RNA, etc., are presented.

### 4.1. Graphene and Its Derivatives-Based FRET Biosensors

The process when a donor fluorophore in the excited state non-radiatively transfers its own energy to a neighboring acceptor fluorophore so that the acceptor emits fluorescence at its characteristic wavelength is known as FRET. The spectral overlap, distance and relative orientation between the donor and the receptor all affect the efficiency of FRET. The distance and relative orientation between the donor and receptor are particularly important for the construction of highly sensitive FRET biosensors [138]. The FRET biosensor master contains an acceptor fluorophore, a donor fluorophore, a sensor structural domain that can sense the signal and change its conformation, a ligand structural domain that can bind to the conformation change to bring the acceptor and donor in close proximity to cause FRET, and a linker that connects each structural domain [139]. When there is no interaction between two clusters of fluorescent proteins, the corresponding fluorescence is produced separately and detected. If there is an interaction between the donor fluorescent protein and the acceptor fluorescence, the fluorescence produced by the donor fluorescent protein X will be absorbed by the acceptor fluorescence and the presence of fluorescence will not be detected in the cell. FRET biosensors are based on the registration of the wavelength and intensity of emitted light and rely on the FRET principle [140]; It enables the detection of various bioreceptors by fluorescence on/off and has the advantages of portability, rapidity, low detection limit and high sensitivity [141,142].

In 2008, Swathi et al. calculated that graphene could theoretically quench the fluorescence of dyes by resonant energy transfer [143]. Later, it was further demonstrated that 2D nanomaterials could effectively quench the fluorescence of organic dyes, inorganic quantum dots, and upconversion materials. Based on this unique property, 2D nanomaterials are widely constructed as fluorescent sensors for the detection of metal ions, biomolecules, bacteria, and cancer cells, and used as superquencher agents [141,144]. Shi et al. developed a FRET biosensor for *S. aureus-specific* gene sequence detection based on GQDs and gold nanoparticle (AuNPs) pairs (Figure 11a) [145]; this FRET biosensor is realized by immobilizing the capture probe on GQDs and coupling the reporter probe on AuNPs, both of which form a sandwich structure with the target nucleotide, and the proximity of GQDs and AuNPs triggers the FRET effect for gene sequence detection. The biosensor detects *S. aureus* genes with high fluorescence quench efficiency and low detection limit. As shown in Figure 11b, a fluorescent biosensor for the detection of glutathione (GSH) was proposed by Yan et al. [146]. The sensor is composed of MnO_2_ nanosheets and GQDs. The GQDs are synthesized by the redox of potassium permanganate and ethanol. In the absence of glutathione, the nanosheets extinguished the fluorescence of the quantum dots. However, when glutathione is present, MnO_2_ is reduced to Mn^2+^ ions and the quantum dots regain fluorescence. Glutathione can be detected using this sensor at 150 nM with a linear range of 0.5–100 µM; they also found that the sensor can also be used for the detection of glutathione detergents, inhibitors, and is suitable for the non-toxic intracellular measurement of glutathione. Compared with other glutathione detection methods, the sensor has a large dynamic detection range, is fast and easy to detect, and is environmentally friendly. The anti-BPA aptamer was used to detect bisphenol A (BPA) by binding to GO by Zhu et al. (Figure 11c) [147]. Similar to the detection of glutathione, GO quenched the fluorescence when BPA is absent, and when BPA is present, the interaction between BPA and the aptamer prevents its adsorption to GO, reducing the quenching effect and the fluorescence becoming apparent. To improve the interaction strength of biomolecules with 2D nanomaterials, researchers proposed a method to replace organic fluorescent dyes with quantum dots to achieve enhanced fluorescence signal and system stability [148]. For example, monolayer GQDs doped with boron and nitrogen show high affinity for bound nucleic acids and proper non-covalent binding ability compared to undoped monolayer GQDs. The presence of boron and nitrogen promotes the uptake and release of single-stranded DNA, respectively. Based on these properties the detection of HIV DNA in living cells has been successfully achieved [149]. Li’s team proposed a DNAzyme/graphene hybrid material to construct a fluorescent biosensor using the ability of graphene to non-covalently adsorb DNA and *E. coli*-specific DNA enzymes [150]. Exposure of DNAzyme/graphene composites prepared by non-covalent self-assembly to *E. coli* samples resulted in the release of DNAzyme and significant recovery of fluorescence signal in solution; it was able to detect *E. coli* in real-time, with high sensitivity and selectivity. Ribonuclease A plays an important role in the study of multiple diseases as an effective biomarker for disease therapeutics, which can regulate several physiological processes. By combining a fluorescent substrate with rGO, Tong et al. proposed a simple, sensitive, and low-cost biosensor for the quantitative analysis of ribonuclease and targeted drug screening (Figure 11d) [151]. Gao’s team developed a low background, high signal-to-noise ratio graphene nanosheet as a platform to develop a multilayer multiplexed DNA biosensor for the first time [152]; it shows a unique fluorescence quench capability with different affinity for single-stranded DNA and double-stranded DNA for real-time detection of different DNA.

### 4.2. Graphene and Its Derivatives-Based SERS Biosensors

SERS is achieved by providing highly enhanced Raman signals from Raman-active analytes adsorbed on the surface of nanostructures for target detection [153]. Most SERS processes are electromagnetic enhancement mechanisms, where the enhancement is caused by light amplification from localized surface plasmon resonance (LSPR) excitation [154]. Plasma materials are conventional precious and found metals (e.g., silver, gold, copper) with nanoscale characteristics that can be excited by far-field incident light to LSPR, which focuses light on nanoscale edges, tips, or gaps, increasing the local electromagnetic field intensity by 2~5 orders of magnitude, and the SERS signal can reach 10^13^–10^15^ times [155]. Significant initial SERS signals are obtained using electromagnetic enhancement between graphene and its derivatives. The aptamer connecting the two can simultaneously act as a recognition molecule to react with the target to be detected by causing a weakening of the SERS signal, the intensity of which is inversely related to the concentration of the target. Due to its ultra-high sensitivity, SERS is considered as a powerful technique for chemical/biomolecular analysis [156,157]. Depending on the electromagnetic mechanism, precious metal nanostructures (e.g., gold, silver, and copper) have been shown to be effective substrates for obtaining strong Raman enhancement.

In 2009, Liu et al. first reported the use of graphene as a Raman-enhanced SERS substrate [158], initiating the rapid development of 2D nanomaterial-based SERS biosensors. Li et al. examined antibiotics based on the polarity of SERS. Graphene/silver composites deposited on screen-printed electrodes were used to detect antibiotics (Figure 12a) [159]. Different antibiotics are first ionized, and river water samples added with different antibiotics are detected using this sensor, and different antibiotics can be distinguished by the difference in Raman peaks. 2D nanomaterials have a large number of biomolecular attachment sites due to their large specific surface area. In addition, the surface functional groups of 2D nanomaterials are easily modified by noble metal nanostructures with high enhancement factors; 2D nanocomposites make them ideal substrates for high-sensitivity SERS. For example, Liang et al. prepared a SERS biosensor based on GO/Ag nanocomposite to detect _L_-theanine (Figure 12b) [160]; this GO/Ag nanocomposite has the advantage of higher SERS activity and better homogeneity, and the SERS performance is dependent on the size and density of AgNPs, which can generate more SERS “hot spots” for the detection of _L_-theanine down to 10^−7^ M. Yang et al. successfully detected prostate-specific antigen (PSA) by depositing nanosilver on the surface of GO with dual signal amplification of enzymes and nanocomposites [161]; this SERS biosensor has good detection in the range of prostate-specific antigen concentrations from 0.5 pg mL^−1^~500 pg mL^−1^ with a detection limit of 0.23 pg mL^−1^. Li’s group proposed a paper-based SERS biosensor that combines the high sensitivity of SERS detection of graphene oxide-plasmonic gold nanostar hybrids, the enrichment of serum bilirubin and the fluorescence super quench ability, and is able to perform sensitive detection of serum bilirubin with a low detection limit of 0.436 μM (Figure 12c) [162]; this paper-based SERS biosensor is clinically valid and applicable for accurate diagnosis of jaundice and its related diseases. Misuse of antibiotics and bactericides can lead to bacterial resistance to antibiotics. To solve this problem, Ko et al. firstly compounded three materials, Ag/ZnO/rGO, as a multifunctional biosensor for SERS detection and multiplex killing of *E. coli* [163]. The sensor organically combines the photocatalytic properties, photothermal conversion properties, bactericidal ability, and SERS properties of the materials, which can be used for both SERS detection and efficient *E. coli* elimination. Highly sensitive biosensors play an important role in biomedical diagnosis, and some diseases can be diagnosed therapeutically by analyzing human secretions, which have low concentrations of target analytes, and it is significant to fabricate ultra-sensitive biosensors. Guo’s team used a two-step in situ reduction method to prepare Ag-Cu_2_O/rGO nanocomposites as SERS biosensors (Figure 12d) [164]. The sensor is highly sensitive to glucose and uses the glucose level in the fingerprint to distinguish diabetics from normal individuals. Khalil et al. demonstrated a dual-platform SERS-based biosensor for efficient and sensitive DNA detection [165]. The sensor produces a unique enhanced SERS signal by hybridizing DNA sequences joining two SERS platforms using very short probes. The biosensor has a detection limit of 10 fM and can distinguish target sequence differences by single nucleotide variants.

### 4.3. Graphene and Its Derivatives-Based SPR Biosensors

When a metal (or other conductive material) is illuminated by polarized light at a specific angle at the interface of two media (usually glass and liquid), the refracted light disappears completely, leaving only the reflected light, a phenomenon called total internal reflection. When the total internal reflection of light occurs at the interface between glass and metal film, the vanishing wave that penetrates into the metal film will trigger the oscillation of free electrons in the metal to generate surface plasma excitations to form the SPR phenomenon. Signal detection can be achieved by measuring the change in reflectance, angle or wavelength over time. The SPR phenomenon results in a direct, markerless and real-time change in the refractive index of the sensor surface, which is proportional to the biomolecule concentration. To measure ligand-analyte interactions, an interacting molecule (e.g., ligand) is immobilized on the sensor surface while another interacting component (e.g., analyte) flows across the surface and binds to the ligand [166]. If the biological or chemical properties of the sample change, then this will lead to a change in refractive index, which will result in a change in wavelength and angle, so that the change in sample properties can be detected by the change in wavelength and angle.

2D nanomaterials, with their large specific surface area and unique structure, greatly enhance the ability to adsorb biomolecules, and the sensitivity of SPR biosensors is further improved. An SPR sensor for human immunoglobulin antibody detection in serum was proposed by Wu et al. By combining GO immobilized with antibodies with Fe_3_O_4_ nanoparticles, the sensor limit is able to go to 1.88 ng/mL, which is one-hundredth of the conventional sandwich structure SPR sensor [167]. Wang et al. demonstrated an SPR sensor using GO modified by AuNPs for the detection of microRNA (miRNA)-141. The sensor uses DNA strands as biorecognition elements and is able to detect microRNA-141 down to picomolar levels. Compared to other types of miRNAs (miRNA-200a and miRNA-429), the sensor was 5-fold and 8-fold more responsive to miRNA-141, respectively. In addition, the performance of the sensor is 1000-fold higher than that of AuNP alone, Moreover, the sensor is easy to operate, low cost of detection and is non-toxic (Figure 13a) [168]. Zhou et al. reported an SPR sensor using a fiber optic probe coated with rGO/AgNP/AuNP for the detection of *E. coli* (Figure 13b) [169]. For better identification of *E. coli*, the antimicrobial peptide Magainin I-C was deposited on the surface. The sensor has a wide linear detection range, better detection limits than other assays using only a single antibody, a faster detection system, lower cost of detection and better stability. The sensor has a wide linear detection range, better detection limits than other assays using only a single antibody, a faster detection system, lower cost of detection and better stability. The sensor was tested in different liquids (e.g., tap water, apple juice, orange juice) with good sample recoveries, proving that the sensor is also widely used in real samples. Omar et al. investigated an SPR sensor based on self-assembled monolayer/rGO-polyamide amine dendrimer films for the detection of dengue virus type 2 E-proteins (DENV-2 E-proteins) [170]. Biomolecular detection is achieved by monitoring the plasma signal changes caused by antibody-antigen affinity reactions [171,172]; this method of covalently bonding antibodies using graphene surfaces provides extremely low detection limits. Gong et al. proposed a d-shaped plastic fiber SPR biosensor based on graphene/gold thin film for detecting the DNA hybridization process. The Au film was grown directly on graphene on copper foil using physical vapor deposition, and the graphene and gold nanofilms achieved seamless contact, which effectively improved the sensitivity of the SPR sensor [173]. Kushwaha et al. developed a ZnO, Au, and graphene-based SPR biosensor for the detection of pseudomonas-like bacteria. A hybrid structure of ZnO, Au, and graphene was coated on a prismatic basis as an affinity layer, and water was used as the sensing medium [174]. The molecular recognition sites on graphene were tightly bound to pseudomonas-like bacteria, ZnO changed the resonance angle, and the hybrid structure significantly improved the accuracy and sensitivity of this SPR biosensor for pseudomonas-like bacteria. With its miniaturization and flexibility, fiber-based technology makes it possible to implant SPR sensors into the human body to detect glucose. However, due to the miniaturization of fiber optic SPR sensors, their sensing area is limited, and their sensitivity is low compared to conventional prismatic SPR sensors. Yu et al. proposed a d-type fiber-optic SPR sensor using CVD to modify graphene and MoS_2_ on the sensor surface to obtain MoS_2_-graphene composite nanostructures (Figure 13c) [175]. The specific receptor pyrene-1-boronic acid (PBA) was modified on graphene to achieve specific detection of glucose, and the sensitivity of glucose detection was up to 6708.87 nm/RIU. Singh et al. developed two different tapered fiber optic sensing probes to detect uric acid [176]. GO and AuNPs were wrapped around the two fiber optic probes to increase the sensing area, excite surface plasmon excitations, and sense refractive index changes around the sensing head using localized surface plasmon resonance phenomena. Due to the presence of uricase, the sensor probe is highly selective for uric acid and has a wide detection range with good performance in the linear range of 10 pM~800 μM.

### 4.4. Graphene and Its Derivatives-Based Colorimetric Biosensors

Among optical biosensors, colorimetric biosensors are capable of producing signals visible to the naked eye that can be observed without the use of other tools. Colorimetric sensing is closely related to the materials used to make the sensors. Depending on the sensing material, the sensing mechanism of the colorimetric sensor varies [177]. Colorimetric sensing of metal nanomaterials is usually achieved based on changes in the optical properties of different SPRs when metal nanoparticles are aggregated and dispersed and monodispersed and aggregated nanoparticles exhibit different colors that can be determined by the naked eye [178]. Enzyme-catalyzed colorimetric sensors are usually based on the color change caused by enzyme catalysis of 3,3,5,5-tetramethylbenzyl to achieve sensing [179]. The different types of enzymes can be divided into natural enzymes and mimetic enzymes. The principle of fluorescence-switched colorimetric sensors is usually based on the fluorescence properties of certain materials [180]. For example, the addition of a burst dye fluorescent analyte can change the fluorescent chromophore from “on” to “off”. Secondly, the induced emission of the fluorescent polymer allows for a color change from “off” to “on”. The emission spectrum of the indicator can be changed by the interaction between the ligand and the receptor, which can also provide a colorimetric response to different concentrations of the analyte [181]; it has the advantages of fast and accurate detection and low detection cost [182,183]. Colorimetric biosensors enable detection by using enzymes or enzyme-like materials to catalyze the chemical reactions that accompany color changes.

GO is used in colorimetric biosensing systems by virtue of its inherent peroxidase activity and its high stability [184]. Huang et al. proposed a colorimetric immunoassay for the highly sensitive detection of respiratory syncytial virus (RSV) based on the peroxidase-like activity of AuNPs-GO hybrids with detection limits as low as 0.04 pg mL^−1^ [185]. Guo et al. demonstrated a simple wet chemical strategy to prepare synthetic Hessian-graphene hybrid nanosheets (H-GNs). H-GNs not only possess intrinsic peroxidase-like activity, reacting with tetramethylbenzidine (TMB) and hydrogen peroxide to produce colored compounds, but also distinguish ss-DNA from ds-DNA at optimal electrolyte concentrations due to the different affinities of ss-DNA and ds-DNA to H-GNs. Based on these unique properties of H-GNs, a colorimetric detection system was developed for the detection of single nucleotide polymorphisms (SNPs) in DNA (Figure 14a) [186]. MicroRNAs (miRNAs), have become biomarkers for a variety of diseases. However, traditional methods to detect miRNAs are complicated and costly, limiting practical clinical applications. Lee and colleagues proposed a new graphene oxide (GO)-based NET strategy for simple miRNA detection with higher sensitivity (Figure 14b) [187]. The method is based on a simulated peroxidase DNAzyme (Dz)-mediated colorimetric assay that uses a bimolecular beacon to exchange strands with miRNAs to improve the color change of colorimetric detection. Sharma et al. prepared a CuO:GNS nanocomposite by hybridizing Cuo with graphene for the colorimetric detection of H_2_O_2_ and cholesterol. The sensor exhibited good sensitivity and selectivity for cholesterol (LOD of 78 pM) and H_2_O_2_ (LOD of 6.88 μM) with high stability at different temperatures. The biocompatibility of the composite CuO:GNS is superior to that of CuO (Figure 14c) [188]. Rostami established a GNR/AgNPs based colorimetric sensor for the detection of dopamine (DA) and GSH in human serum by hybridizing graphene nanoribbons/silver nanoparticles (GNR/AgNPs) using the plasma hybridization technique (Figure 14d) [189]. DA was detected based on etching and transformation of the morphology of GNR/AgNPs, which showed a blue shift in its plasma band during etching, along with a color change from green to red. Subsequently, glutathione induced aggregation of AgNPs, resulting in a decrease in the absorption intensity of the AgNPs plasma band and a color change from red to gray for glutathione detection. Compared with the colorimetric sensor without the addition of GNR, the detection limit was lower by using GNR/AgNPs hybridization as the colorimetric sensor.

In this section, we summarize the discussion of optical biosensors based on graphene and its derivatives, mainly including FRET biosensors, SERS biosensors, SPR biosensors, and colorimetric biosensors. Although these optical biosensors work on different principles, they are all capable of being able to accurately identify biomolecules such as DNA, RNA, proteins, glucose, bacteria, viruses, etc., with low detection limits and wide detection ranges, and present them in Table 2.

Graphene nanomaterials with a high surface-to-volume ratio and excellent FRET-based distance-dependent fluorescence bursting ability are ideal materials for modifying fluorescent biosensors [190]. Although the complete fluorescence mechanism of graphene and its derivative materials is not known, several types of graphene derivative materials such as graphene, GO, rGO, and GQDs have been used for fluorescent biosensors [191,192]. Graphene and graphene oxide are ideal fluorescence quenchers due to their large electron planes on nanosheets, which quench fluorescence by FRET [193]. In addition, multicolor GQDs provide powerful fluorescence, making them very useful in optical biosensors [194].

Theory and experiments have shown that graphene and its derivatives can efficiently excite and propagate surface plasma excitations [195]. Although graphene and its derivative materials can significantly improve the sensitivity of SPR biosensors due to their extraordinary optical absorption and plasma properties, they tend to adversely reduce the signal-to-noise ratio and quality factor of the biosensor [196]. The synthesis of new graphene-based nanomaterials by surface coating or functionalization can effectively eliminate these defects and maintain or even enhance their sensing properties [197].

Graphene and graphene derivatives, such as GO and rGO, are favored due to their stronger SERS effects. The observed enhancement of the SERS signal is a chemically enhanced process due to the charge transfer between the graphene substrate and the adsorbed molecules [198]. GO exhibits several unique Raman scattering features, such as high-frequency D (disorder) and tangential modes (g-band), which are distinct, obvious, and easily distinguished from the fluorescence background and are good reporters of the Raman signal [199]. However, the Raman signal of GO is weak and can hardly be used for Raman detection, while the Raman signal of GO can be significantly enhanced by combining it with noble metal nanoparticles through non-covalent functionalization [200].

Graphene and its derivatives can provide more active sites for biosensing due to their high specific surface area properties, thus improving the sensitivity of the prepared colorimetric biosensors [201]. Graphene and its derivatives can be used as an effective linking platform to couple with nanoparticles, quantum dots, polymers, and biomolecules to form functional hybrid nanomaterials, which can improve the sensitivity and selectivity of the prepared colorimetric sensors [202].

## 5. Conclusions

Graphene and its derivatives are a new class of two-dimensional materials with unique structures and superior properties, which are widely used in various fields, including energy storage, superconductors, electronic devices, and biosensing. Although electrochemical sensors based on graphene and its derivatives have been widely used in a biosensor, they can only measure the change of current on the surface, which may cause damage to living cells and affect the test results. However, graphene-based optical sensors enable biomolecule detection through fluorescence, Raman spectroscopy, and refractive index changes induced by target molecules; these optical biosensors have a wide detection range, high detection accuracy, and accurate and rapid recognition, and have potential applications in biosensing; this paper focuses on the preparation processes (such as CVD, epitaxial growth, chemical reagent oxidation, hydrothermal method, etc.) of graphene and its derivatives (graphene oxide, reduced graphene oxide, graphene quantum dots). We introduce optical biosensors for biomolecule detection: FRET biosensors, SERS biosensors, SPR biosensors, colorimetric biosensors, and summarize the research progress of graphene and its derivatives in the field of sensing. Graphene and its derivatives are two-dimensional layered materials composed of sp^2^ atoms, which have a large specific surface area and can provide a large number of attachment sites for biomolecular detection. Moreover, GO, rGO, and GQDs have oxygen-containing functional groups on their surfaces, which have been chemically modified and can be dispersed in various solvents with good biocompatibility. In particular, graphene and its derivatives possess unique optical properties, such as broadband and tunable absorption, fluorescence quenching, and strong polarization-dependent effects; these excellent properties make it possible to achieve biomolecular detection non-destructively. The structures and properties of graphene and its derivatives for these sensors are investigated, according to which their structural properties can be used in biosensing to achieve high sensitivity, selectivity and reproducibility applications; this paper provides a detailed review of the recent research progress in the field of graphene and its derivatives optical sensors for biomolecule detection. Optical biosensors based on graphene and its derivatives can not only detect cytogenetic material as well as metabolites, but also have a high level of detection sensitivity and accuracy. With the advantages of wide detection range, high detection sensitivity and accurate and rapid identification of target cells, these optical biosensors are expected to be widely used in cellular biomedical research for anti-cancer drug screening, disease treatment and detection of unlabeled living cells.

## 6. Challenges and Opportunities

Novel optical biosensors and other applications using graphene and its derivatives as sensing elements have been developed, which can be used for the detection of a range of biomolecules such as single cells, cellular secretions, proteins, nucleic acids, and antigen-antibodies; these new high-performance optical sensors are capable of detecting changes in surface structure and biomolecular interactions with the advantages of ultra-fast detection, high sensitivity, label-free, specific recognition, and the ability to respond in real-time. However, the application areas of graphene and its derivatives-based biosensors are limited, and many applications are restricted to biomolecule detection, which needs to be extended to other medical fields. In addition, efforts should be made to achieve the applicability, reusability, and low cost of new biosensors to adapt to various complex environments, not limited to the laboratory, and to achieve scale-up production, which still faces serious challenges.

(1)Understanding the interaction mechanism between biomolecules and graphene and its derivative materials. The analytical performance of optical biosensors based on graphene and its derivatives is linked to the number and arrangement of biomolecules on the surface of graphene and its derivative materials.(2)The homogeneity of graphene and its derivatives in terms of size, thickness, biocompatibility, and the number of surface functional groups is difficult to ensure; this inhomogeneity has a great impact on the performance evaluation and reproducibility of the constructed optical biosensors. Optimization of production methods and improved control of the size and morphology of the produced particles and standardization of the manufacturing process.(3)The type and functionalization of precursors during the preparation of graphene and its derivatives often introduce different impurities, and further research should be conducted to simplify effective sample preparation methods to avoid the interference of other substrates and to ensure that their biosensing performance as signal nanoprobes is not affected.(4)It is also a serious challenge to develop a biofunctionalization method that maintains high stability without changing the structure and function of biomolecules and graphene and its derivative materials, to develop biosensor devices with high stability and reproducibility, and to expand the practical applications of biosensors based on graphene and its derivative materials.(5)Emphasis should be placed on miniaturization, standardization, and multiple readouts of biosensors to reduce the incidence of false negative/positive results. Further miniaturization, integration with fluidic systems, and integration of multi-chip systems are important challenges for the next generation of graphene and its derivatives optical biosensors.(6)The combination of graphene and its derivative optical biosensors with smartphones, in particular, has great potential to enable low-cost, reliable and versatile biosensing platforms that can be widely used in the field and provide excellent convenience for on-site detection of pathogens in remote areas.

## Figures and Tables

**Figure 1 ijms-23-10838-f001:**
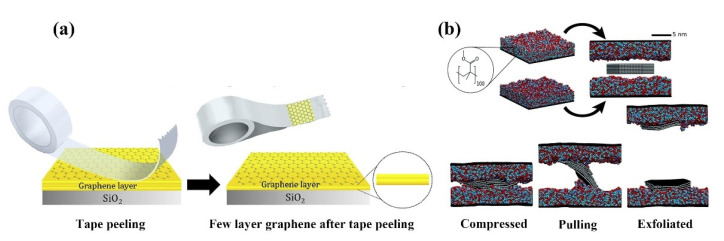
(**a**) Schematic diagram of graphene preparation on SiO_2_ using tape-peeling method. Reprinted with permission from Ref. [45]. Copyright 2020, Elsevier. (**b**) Schematic diagram of graphene exfoliation using polymeric tape. Graphite is first compressed between two polymer layers, and then the interval between the polymer layers is increased to exfoliate the graphite to prepare graphene. Reprinted with permission from Ref. [46]. Copyright 2019, Royal Society of Chemistry.

**Figure 2 ijms-23-10838-f002:**
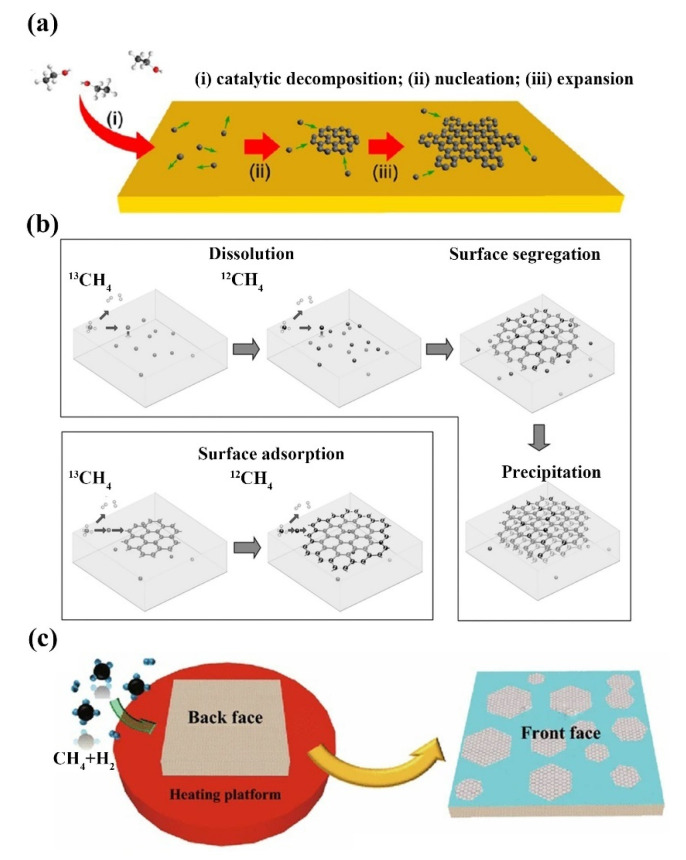
(**a**) Schematic of the initial state of the growth of graphene on copper from ethanol. Reprinted with permission from Ref. [51]. Copyright 2013, American Chemical Society. (**b**) Schematic diagram of the possible distribution of C isotopes in graphene films based on different growth mechanisms of CVD (surface segregation/surface adsorption). Reprinted with permission from Ref. [52]. Copyright 2016, Wiley Online Library. (**c**) Schematic of graphene growing on an inverted silicon surface. Reprinted with permission from Ref. [56]. Copyright 2018, Springer Nature.

**Figure 3 ijms-23-10838-f003:**
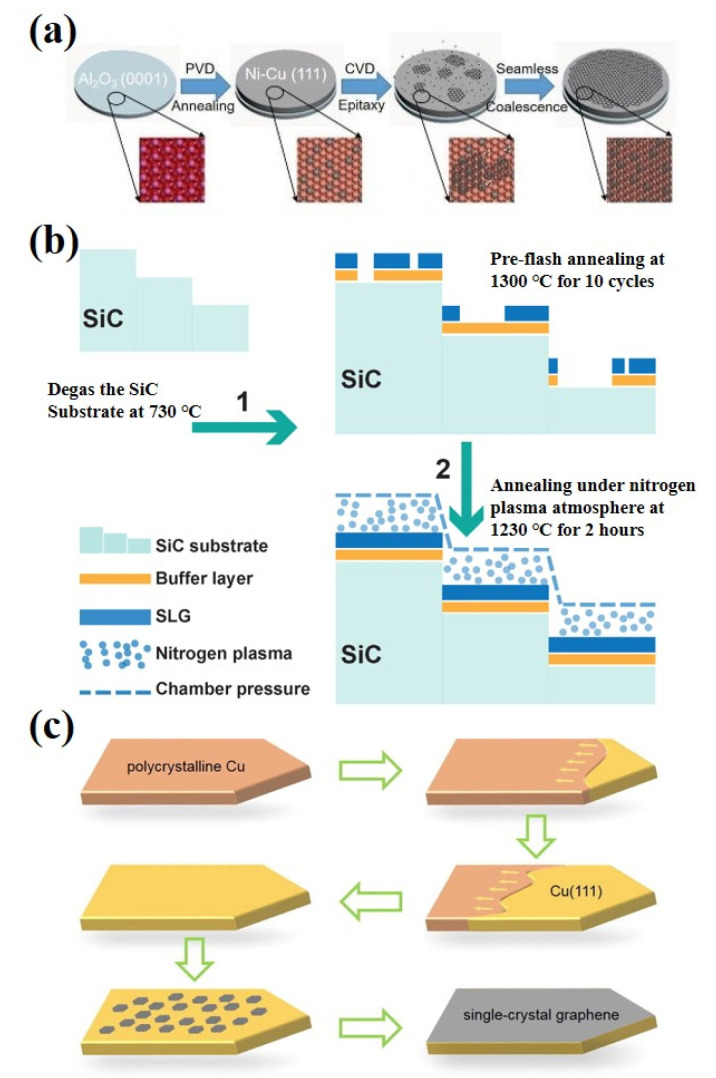
(**a**) Schematics for the fabrication of wafer-scale single-crystal graphene on Cu/Ni (111) alloy film at low temperature. Reprinted with permission from Ref. [62]. Copyright 2019, Wiley Online Library. (**b**) Schematic diagram of preparing uniform SLG on 4H-SiC (0001). The figure involves three main processes: the degassing of the substrate, pre-flash annealing, and annealing under a nitrogen plasma atmosphere. Reprinted with permission from Ref. [63]. Copyright 2021, MDPI. (**c**) Schematic diagram of ultrafast epitaxial growth of large-size graphene films. Reprinted with permission from Ref. [64]. Copyright 2017, Elsevier.

**Figure 4 ijms-23-10838-f004:**
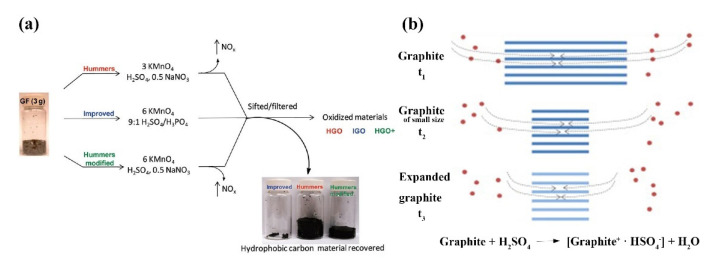
(**a**) Graphite flakes (GF) were used as raw material to prepare improved GO (IGO), Hummers GO (HGO), and Hummers modified (HGO^+^) using three different methods. Reprinted with permission from Ref. [71]. Copyright 2010, Elsevier. (**b**) Schematic diagram of GO preparation by spontaneous expansion using expanded graphite as a precursor. Reprinted with permission from Ref. [72]. Copyright 2013, Elsevier.

**Figure 5 ijms-23-10838-f005:**
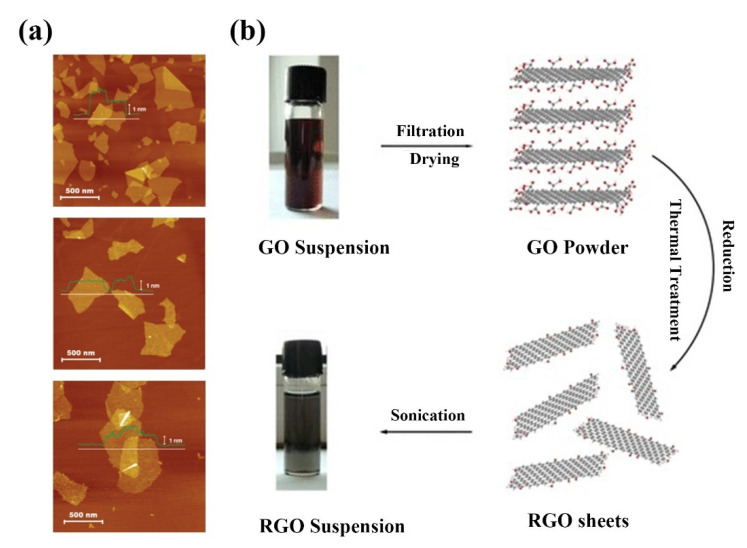
(**a**) AFM images of unreduced (**top**), hydrazine-reduced (**middle**), and vitamin c-reduced (**bottom**) GO flakes deposited from the corresponding suspensions onto the substrate. Reprinted with permission from Ref. [82]. Copyright 2010, American Chemical Society. (**b**) Schematic diagram of rGO suspension prepared by the low-temperature thermal treatment. Reprinted with permission from Ref. [85]. Copyright 2017, Springer Nature.

**Figure 6 ijms-23-10838-f006:**
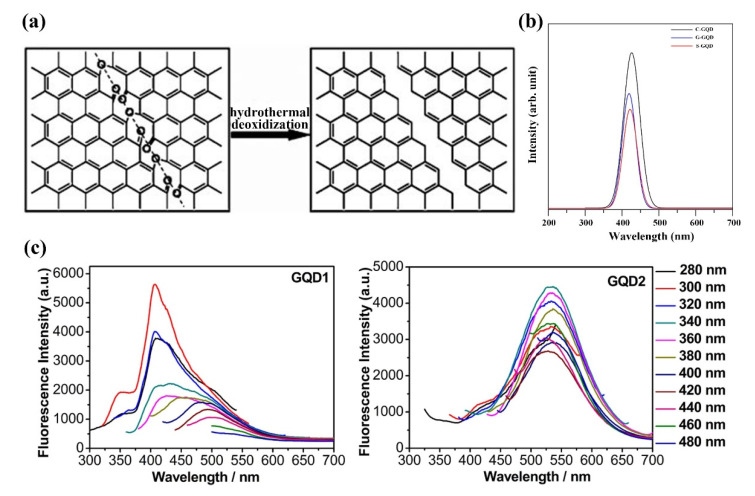
(**a**) Mechanism for the hydrothermal cutting of oxidized GSs into GQDs. Reprinted with permission from Ref. [94]. Copyright 2010, Wiley Online Library. (**b**) PL spectra of different GQDs prepared by CNT, SP and GO, precursors. Reprinted with permission from Ref. [97]. Copyright 2017, Elsevier. (**c**) Fluorescence spectra of GQD1 (**left**) and GQD2 (**right**) under different excitation wavelengths. Reprinted with permission from Ref. [98]. Copyright 2018, MDPI.

**Figure 7 ijms-23-10838-f007:**
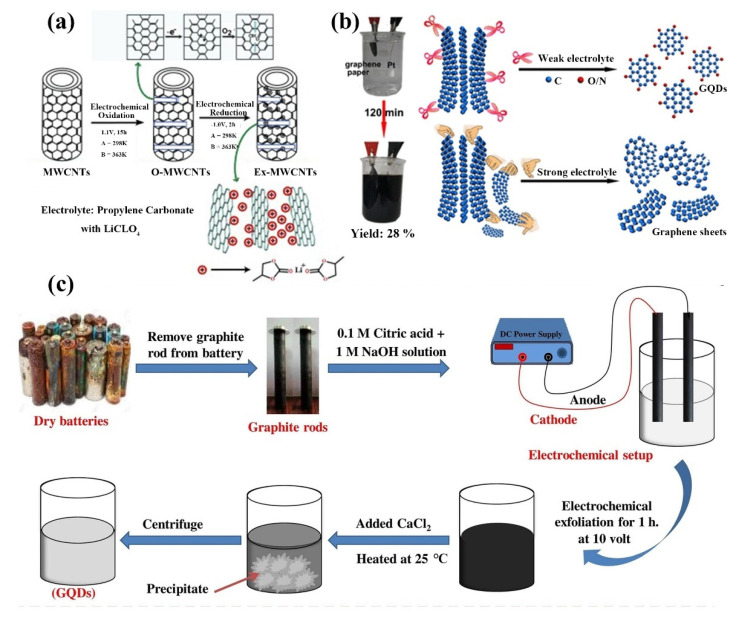
(**a**) Green luminescent GQDs with diameters of 3.5 and 8.2 (±0.3) nm, respectively, were prepared electrochemically at 90 °C using LiClO_4_, while similar particles with diameters of 23 (±2) nm were obtained at 30 °C under the same conditions. Reprinted with permission from Ref. [102]. Copyright 2012, Wiley Online Library. (**b**) A schematic diagram of the preparation of GQDs in strong and weak electrolyte solutions. Reprinted with permission from Ref. [106]. Copyright 2018, American Chemical Society. (**c**) The diagram of electrophoretic exfoliation to produce GQDs. Reprinted with permission from Ref. [107]. Copyright 2021, Elsevier.

**Figure 8 ijms-23-10838-f008:**
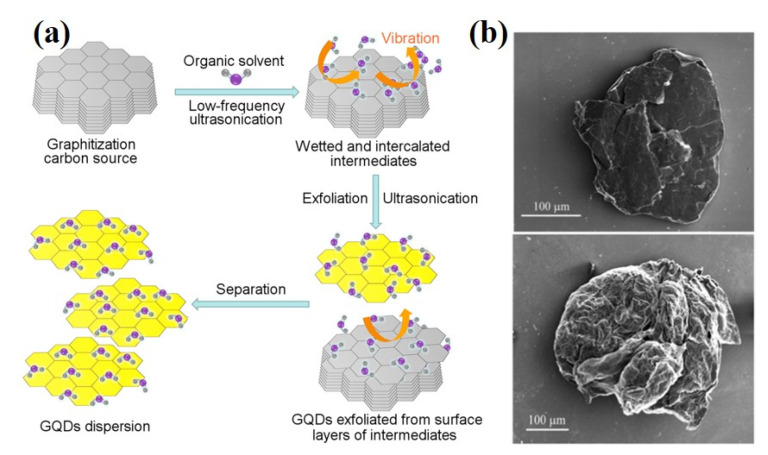
(**a**) Schematic illustration of the preparative process of GQDs by ultrasonic-assisted exfoliating graphitic carbon precursors (acetylene black, nano-graphite) in organic solvents. Reprinted with permission from Ref. [109]. Copyright 2016, Elsevier. (**b**) SEM images comparing graphite before (**top**) and after (**bottom**) the microwave expansion. Reprinted with permission from Ref. [110]. Copyright 2018, Elsevier.

**Figure 9 ijms-23-10838-f009:**
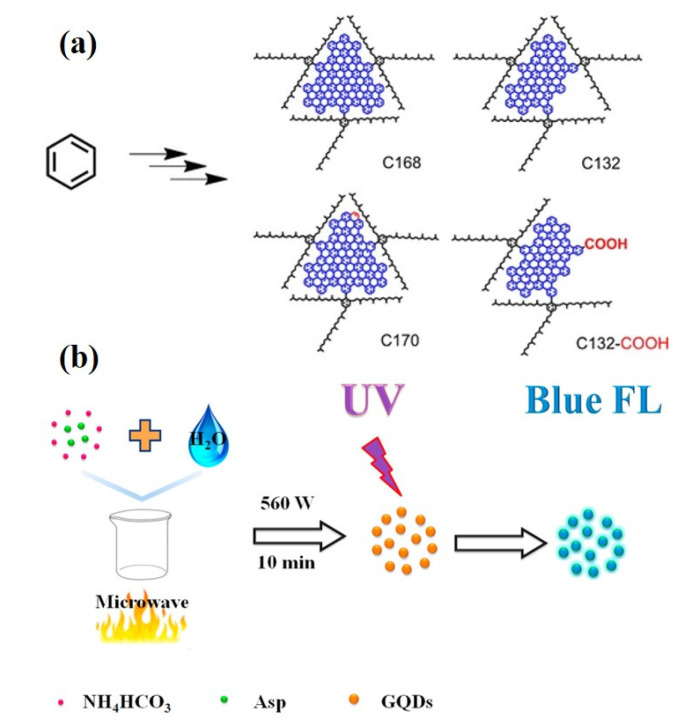
(**a**) Stepwise Organic Methods: schematic structure of colloidal graphene quantum dots synthesized by stepwise solution chemistry control with a graphene core containing 168, 132, 170 and 132 conjugated carbon atoms. Reprinted with permission from Ref. [113]. Copyright 2013, American Chemical Society. (**b**) Microwave-Assisted Methods: schematic illustration of the preparation process for the GQDs. Reprinted with permission from Ref. [116]. Copyright 2016, Elsevier.

**Figure 10 ijms-23-10838-f010:**
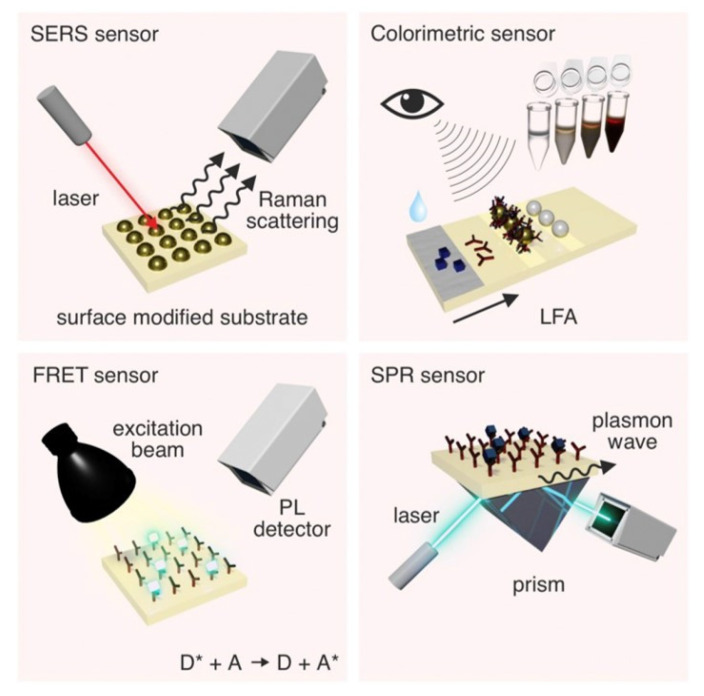
Overview of graphene-based optical sensors. Reprinted with permission from Ref. [133]. Copyright 2020, Elsevier. The * represents the energy transferred from the donor fluorescent protein to the acceptor fluorescent protein.

**Figure 11 ijms-23-10838-f011:**
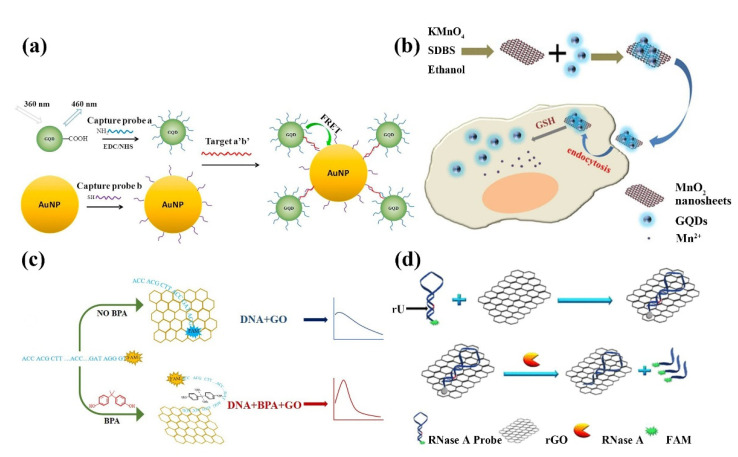
(**a**) Schematic illustration of a FRET assay or the detection of Staphylococcus *S. aureus* gene between graphene quantum dots and gold nanoparticles. Reprinted with permission from Ref. [145]. Copyright 2015, Elsevier. (**b**) Activation mechanism of the MnO_2_ nanosheet-GQDs nanoprobe for GSH detection in vitro. Reprinted with permission from Ref. [146]. Copyright 2016, American Chemical Society. (**c**) Schematic illustration of the biosensor for BPA based on the target-induced conformational change of the anti-BPA aptamer and the interactions between the FAM-ssDNA probe and GO. Reprinted with permission from Ref. [147]. Copyright 2015, American Chemical Society. (**d**) Schematic illustration of RNase A assay. Reprinted with permission from Ref. [151]. Copyright 2019, American Chemical Society.

**Figure 12 ijms-23-10838-f012:**
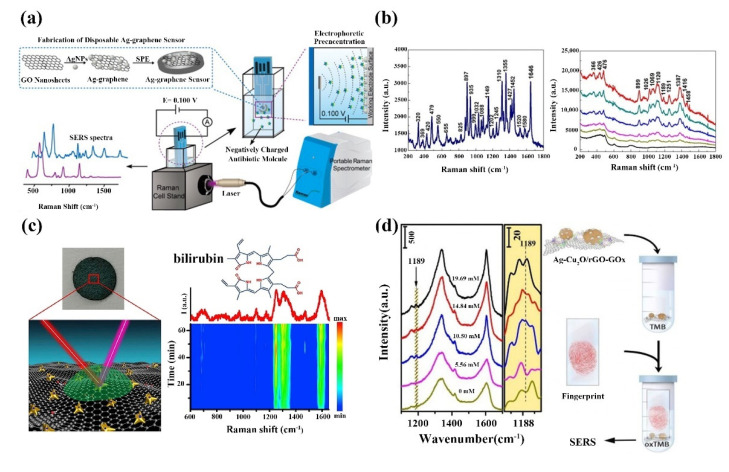
(**a**) Schematic diagram of a disposable ag-graphene sensor for the detection of polar antibiotics in water. Reprinted with permission from Ref. [159]. Copyright 2013, Elsevier. (**b**) Normal Raman spectra of _L_-theanine (**left**); SERS spectra (**right**) of _L_-theanine adsorbed on GO/silver nanocomposites at different concentrations (10^−2^, 10^−3^, 10^−4^, 10^−5^, 10^−6^, 10^−7^ mol/L from top to bottom, respectively). Reprinted with permission from Ref. [160]. Copyright 2017, Elsevier. (**c**) Schematic illustrating the operating principle (**left**) of the developed enPSERS biosensor, molecular structure (**top right**) and the averaged SERS spectrum (**bottom right**) over the measurement period of bilirubin. Reprinted with permission from Ref. [162]. Copyright 2019, Elsevier. (**d**) SERS spectra (**left**) of oxidized tetramethylbenzidine (TMB) molecules on the surface of fingerprints of diabetic patients and normal subjects in the presence of Ag-Cu_2_O/rGO substrate and glucose oxidase (Gox). A schematic diagram (**right**) of SERS detection of fingerprints using Ag-Cu_2_O/TGO nanocomposites as SERS substrates. Reprinted with permission from Ref. [164]. Copyright 2017, American Chemical Society.

**Figure 13 ijms-23-10838-f013:**
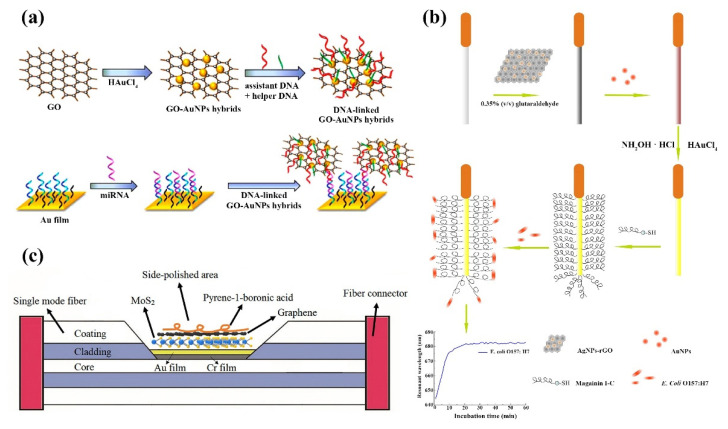
(**a**) Schematic diagram of the GO-Au-SPR biosensor process. Reprinted with permission from Ref. [168]. Copyright 2016, Elsevier. (**b**) Magainin I-C immobilized on gold-coated optical fiber by Au-S bonding can specifically capture *E. coli* O157:H7, resulting in the change of refractive index of the fiber surface and the wavelength shift of the surface plasma absorption peak, enabling the qualitative and quantitative detection of E. coli O157:H7. Reprinted with permission from Ref. [169]. Copyright 2018, Elsevier. (**c**) A d-type fiber-optic SPR sensor for glucose detection was prepared using MoS_2_-graphene composite nanostructures and PBA. Reprinted with permission from Ref. [175]. Copyright 2020, Elsevier.

**Figure 14 ijms-23-10838-f014:**
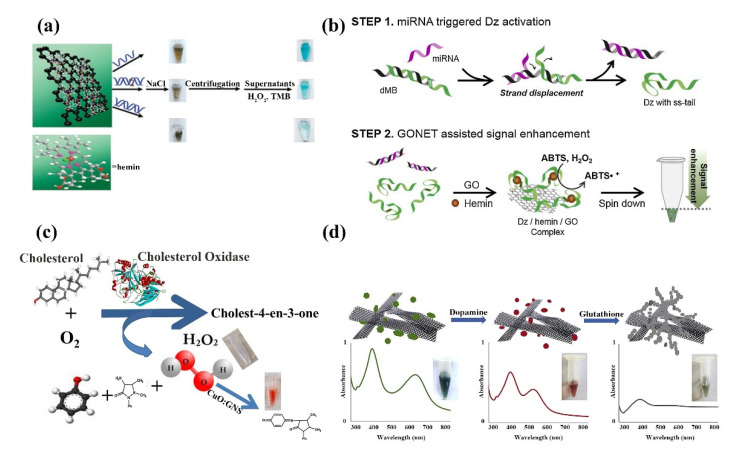
(**a**) Label-free colorimetric detection system for the detection of single nucleotide polymorphisms (SNPs) in disease-associated DNA. From top to bottom, they are ssDNA, single misaligned double-stranded DNA and complementary double-stranded DNA. Reprinted with permission from Ref. [186]. Copyright 2011, American Chemical Society. (**b**) Schematic diagram of GONET-based advanced colorimetric miRNA detection. Reprinted with permission from Ref. [187]. Copyright 2019, Elsevier. (**c**) Schematic diagram of the process of cholesterol detection by colorimetric method. Reprinted with permission from Ref. [188]. Copyright 2017, Elsevier. (**d**) Colorimetric sensors based on graphene/silver hybrid materials for the detection of dopamine (DA) and GSH. Reprinted with permission from Ref. [189]. Copyright 2020, Elsevier.

**Table 1 ijms-23-10838-t001:** Preparation methods of graphene and its derivatives and their advantages and disadvantages.

Materials	Methods	Advantages	Disadvantages	Ref.
Graphene	Mechanical exfoliation	very simple process, large-scale production, good structural and electronic quality	low yields, time-consuming	[45]
	CVD	uniform, high-quality, large-scale production	high cost, complicated technique, high temperature requirement	[51]
	Epitaxial growth	Wafer-scale production, high qualities	low yields, very high temperature requirement, high synthesis cost	[63]
GO	Boride	low-cost, simple	produce harmful gases, low yields	[65]
	Staudenmaier	very simple process, low-cost,	produce harmful gases, low yields, time-consuming	[66]
	Hummer’s	low-cost, simple, environmentally friendly, Fast,	heavy metal pollution, low purity, dangerous	[67]
	Modified hummer’s	environmentally friendly, high purity, high oxidation degree and regular structure	low yields, complicated	[71]
	Microwave radiation	high purity, fast, uniformly sized,	low yields, high temperature required	[73]
	Mild thermal annealing	environmentally friendly, simple	low yields, low purity, time-consuming	[74]
rGO	Hydrazine hydrate reduction	simple, low-cost, high qualities	high toxicity, low yields	[81]
	Vitamin C reduction	environmentally friendly, stable, high purity	high temperature requirement	[82]
	Ethanol vapor reduction	environmentally friendly, high qualities	high temperature requirement	[83]
	Hydroquinone reduction	structure orderly, high purity	easy to agglomerate in water, poor thermal stability	[84]
	Thermal reduction	Rapidly, high qualities, low-cost	high temperature requirement, complicated technique	[86]
	Ultraviolet radiation reduction	large-scale, stable, simple	require very specific reaction conditions,	[90]
	Microwave-assisted reduction	rapidly, uniformly sized, environmentally friendly	low yields, high temperature required	[92]
GQDs	Hydrothermal/solvothermal	Simple, large-scale synthesis	requires higher reaction conditions, Product Random, time-consuming	[94]
	Liquid Phase Exfoliation	low-cost, scalable, high-yield, simple, environmentally friendly	low purity, time-consuming	[100]
	Electrochemical Exfoliation	accurate synthesis, short time consumption, high yields, high crystallinity,	complicated, high cost	[109]
	Cage Opening of Fullerenes	good structural and electronic quality	low yields, complicated, high temperature required, require very specific chemicals and reaction conditions	[111]
	Stepwise Organic	precise control of the structure of the final product	require very specific chemicals and reaction conditions, little potential for scalability	[113]
	Microwave-Assisted	Fast, uniformly sized, size can be easily changed by varying microwave time	low yields, high temperature required	[116]

**Table 2 ijms-23-10838-t002:** Optical biosensors based on graphene and its derivatives.

Types	Biomolecules	Materials	Symbolic Parameters	Ref.
FRET Biosensors	*S.aureus*	GQDs/AuNPs	The limit of detection (LOD) of this FRET biosensor was around 1 nM for *S.aureus* gene detection.	[145]
	GSH	GQDs/MnO_2_	The sensing platform displayed a sensitive response to GSH in the range of 0.5–10 μmol L^–1^, with a detection limit of 150 nmol L^−^^1^.	[146]
	BPA	GO/anti-BPA aptamer	A low limit of detection of 0.05 ng/mL was obtained in the range 0.1–10 ng/mL.	[147]
	HIV DNA	GQDs doped with boron and nitrogen	This DNA sensor shows a linear range from 0 to 20 nM, with a detection limit of 0.5 nM. Moreover, such detection could be finished within 4 min.	[149]
	*E. coli*	DNAzyme/graphene	The sensor is able to detect *E. coli* in this complex matrix with a detection sensitivity of 10^5^ CFU/mL.	[150]
	RNase A	rGO	The method with detection limit of 0.05 ng/mL was first applied for RNase A targeted drug screening.	[151]
SERS Biosensors	Polar antibiotics	Ag–graphene	The detection limits for Methotrexate (MT), 6-aminopenicillanic acid (6-AA), ampicillin trihydrate (AT), and penicillin G (PG), were 0.6, 0.8, 0.7, and 0.3 nM, respectively. Detection could be finished within 10 min.	[159]
	_L-_Theanine	GO/Ag	The detection limit was estimated to be as low as 10^−^^7^ M.	[160]
	PSA	GO/Ag	The SERS immunoassay possesses excellent analytical performance in the range 0.5 pg mL^−^^1^ to 500 pg mL^−^^1^ with a limit of detection of 0.23 pg mL^−^^1^.	[161]
	Serum bilirubin	GO/Au nanostar	The results of SERS detection of bilirubin in blood serum show linear response ranges from 5.0 to 150 μM and 150–500 μM with the detection limit as low as 0.436 μM.	[162]
	*E. coli*	Ag/ZnO/rGO	The LOD was about 10^4^ cfu/mL, and a linear calibration curve in the detectable concentration range of 5 × 10^4^–10^8^ cfu/mL	[163]
	glucose	Ag-Cu_2_O/rGO	Observed for diabetes even with a blood sugar level as low as 10.50 mM.	[164]
	DNA	GO/Au nanoparticle	The biosensors achieve the lowest LOD as low as 10 fM.	[165]
SPR Biosensors	miRNA	GO/AuNPs	The developed SPR biosensor was able to achieve a detection limit as low as 1 fM.	[168]
	*E. coli* O157:H7	antimicrobial peptides/AgNPs-rGO	The SPR biosensor exhibited a good linear relationship with the target bacteria concentration in the range of 1.0 × 10^3^ to 5.0 × 10^7^ cfu/mL with the detection limit of 5.0 × 10^2^ cfu/mL.	[169]
	DENV-2 E-proteins	Self-Assembled Monolayers/rGO-PAMAM Dendrimer	Sensitive detection of DENV-2 E-proteins was performed in the range of 0.08 pM to 0.5 pM.	[170]
	DNA	graphene/Au film	For target DNA with the detection limit of 10^−^^10^ M.	[173]
	pseudomonas like bacteria	ZnO/Au/graphene	The proposed biosensor has a greater sensitivity of 187.43 deg/RIU, detection accuracy of 2.05 deg^−1^ and quality parameter of 29.33 RIU^−1^.	[174]
	glucose	MoS_2_/graphene	A sensitivity of up to 6708.87 nm/RIU was achieved in glucose detection.	[175]
	uric acid	AuNPs/GO	The linearity test was performed by detecting the different concentrations of uric acid solutions in the range of 10–800 µM.	[176]
Colorimetric Biosensors	RSV	AuNPs/GO	This biosensor based on AuNPs–GO hybrids could provide a LOD of 0.04 pg mL^−1^.	[185]
	DNA	H-GNs	The biosensor exhibited linear response of target DNA in the range of 5–100 nM with a limit of detection around 2 nM.	[186]
	miRNA	GO	The biosensor provides clear visualization of the target at the 10^−^^9^ M scale with the naked eye without any complicated amplification steps.	[187]
	cholesterol	CuO/GNS	The nanocomposite sensor has shown excellent detection sensitivity for cholesterol and has demonstrated a linear response in the range of 0.1–0.8 mM with LOD as low as 78 μM.	[188]
	DA/GSH	GNR/AgNPs	DA and GSH were successfully detected in low concentrations of 0.04 μM and 0.23 μM, respectively.	[189]

## Data Availability

No new data were created or analyzed in this study. Data sharing is not applicable to this article.

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
