# Peer review of "Optical Biosensor Based on Graphene and Its Derivatives for Detecting Biomolecules"

_ijms, 2022, doi:10.3390/ijms231810838_

Round 1

Reviewer 1 Report

The paper review the optical biosensors based on graphene derivatives for optical biosensor applications. In addition, the review includes the synthesis or preparation process of the graphene derivatives. The paper provides good coverage and has merit for publication 

but requires further clarification/discussion in the text as follows: 

1.    The quality of the Figures and labels in the Figures need to be improved significantly.

2.    The content is appropriate but lacks deeper discussion, especially in the sensing approach. 

3.    The importance of the concept/principle of FRET, SERS, SPR and colorimetric was not discussed. They should do this before they discuss what is reported in the literature. They should start with a principle of those methods, and the important criteria and then highlight what other has been done to close the gap. 

4.    I can see the discussion on synthesis, fabrication, or preparation processes of graphene and derivative, but I did not see the importance of surface modification or attachment approach/method for those mentioned materials.

5.    The challenge and future perspective of those materials in optical biosensors were not discussed. I can see the challenges/problems in the conclusion, but it is not comprehensive and enough. Expansion discussion in one sub-chapter will significantly improve the paper. 

6.    What factors those materials provide a high level of detection sensitivity and accuracy were not highlighted. 

Author Response

Responses to Reviewer's Comments

  1. The quality of the Figures and labels in the Figures need to be improved significantly.

Response: Thank you for your valuable comment and suggestion. We have modified all the Figures in the article accordingly, and the quality of the Figures and labels in the Figures have been significantly improved.

  1. The content is appropriate but lacks deeper discussion, especially in the sensing approach.

Response: Thank you for your valuable comment and suggestion. We have thought deeply about the sensing method by reviewing the relevant literature, and the relevant contents are added in the revised manuscript.

Correction is marked with red color in line 713, 784, 861, 920 of the revised manuscript.

  1. The importance of the concept/principle of FRET, SERS, SPR and colorimetric was not discussed. They should do this before they discuss what is reported in the literature. They should start with a principle of those methods, and the important criteria and then highlight what other has been done to close the gap.

Response: Thank you for your valuable comment and suggestion. We have thoroughly checked and revised the manuscript and some changes have been made to improve the manuscript.

Correction is marked with red color in line 704 to 718, 780 to 792, 850 to 864, 925 to 940 of the revised manuscript.

  1. I can see the discussion on synthesis, fabrication, or preparation processes of graphene and derivative, but I did not see the importance of surface modification or attachment approach/method for those mentioned materials.

Response: Thank you for your valuable comment and suggestion. By reviewing the relevant literature, we have gained a deeper understanding of surface modification or biomolecular functionalization.

Correction is marked with red color in line 626 to 682 of the revised manuscript.

  1. The challenge and future perspective of those materials in optical biosensors were not discussed. I can see the challenges/problems in the conclusion, but it is not comprehensive and enough. Expansion discussion in one sub-chapter will significantly improve the paper.

Response: Thank you for your valuable comment and suggestion. We have thoroughly checked and revised the manuscript to explore in more depth the challenges that graphene and its derivative optical biosensors may overcome in the future.

Correction is marked with red color in line 1065 to 1093 of the revised manuscript.

  1. What factors those materials provide a high level of detection sensitivity and accuracy were not highlighted.

Response: Thank you for your valuable comment and suggestion. There are many factors that affect sensing sensitivity, the electrostatic, π–π stacking, hydrogen bonding and hydrophobic forces between biomolecules and graphene and its derivative materials play an important role in facilitating their interactions. Since the surfaces of GO and rGO contain oxygen-containing functional groups such as carboxyl and hydroxyl groups, they can effectively immobilize biomolecules containing a variety of functional groups. Alternatively, metal nanoparticles (e.g. AuNPs, AgNPs, PtNPs) can also be used as modifiers of graphene-based nanomaterials (especially GO and rGO) linkers to achieve non-covalent functionalization. These metal nanoparticles can not only achieve stable connections between graphene and biomolecules, but also enhance and amplify sensing signals.

Correction is marked with red color in line 644,662, 670 of the revised manuscript.

Reviewer 2 Report

This paper reviews the optical biosensor based on graphene and its derivatives for non-destructive detection of biomolecules. The preparation of graphene and its derivatives are summarized and recent advancement in the development of graphene-based optical biosensors are discussed. In addition, the challenges in the current stage of application are described and future prospects are presented.  This review needs to be revised carefully and hopefully the comments below will be able to help to further improve the review.

1. An abstract is often presented separately from the article, so it must be able to stand alone. Hence the problem statement, aim, novelty and the recent gap in this field have to be all included.

2. Graphical abstract: Not provided, it will be interesting for the author to present the review through figure for better illustration purpose.

3. Figures need to be supplied at higher resolution.

4. Introduction: The author could elaborate on the importance of optical biosensor and the advantage of its application for biomolecular detection.

5. Introduction: The author could also briefly introduce the different applications of graphene due to its remarkable properties. Kindly refer to the relevant papers:

a. “A proof of concept: detection of avian influenza H5 gene by a graphene-enhanced electrochemical genosensor”

b. “Graphene-based field-effect transistor biosensors for the rapid detection and analysis of viruses: A perspective in view of COVID-19”

c. “Label-free terahertz microfluidic biosensor for sensitive DNA detection using graphene-metasurface hybrid structures”

6. Section 2 is too general and too short to be a standalone section, it is suggested that the author could simplify it and combine it with Section 3.

7. Section 3: The synthesis approach for graphene and its derivatives has been summarized in many previous reviews, the author could provide a brief summary and discuss on the properties of graphene and its derivatives synthesized using different approaches.

8. Subsection titles for Section 4 should be edited accordingly, eg. Section 4.1 should be “graphene and its derivatives-based FRET biosensors”.

9. Line 587: The scientific names of bacteria (genus and species name) should be italicized.

10. The biosensing performance of the recent advancement in the graphene and derivatives-based optical biosensor should be tabulated for easy referencing.

11. It would be good if the author could provide some take home messages after summarizing the works done on the graphene and derivatives-based optical biosensor. For example, which graphene or the derivative is more suitable for a certain type of optical biosensor and what are the advantage and disadvantage, etc.

12. Line 804: The sentence should be changed to “… widely used to modify biosensor/ … widely used in biosensor”.

13. The author needs to discuss more in-depth for the possible future prospect in overcoming the challenges of graphene and derivative-based optical biosensor.

Author Response

Responses to Reviewer's Comments

Reviewer #2: This paper reviews the optical biosensor based on graphene and its derivatives for non-destructive detection of biomolecules. The preparation of graphene and its derivatives are summarized and recent advancement in the development of graphene-based optical biosensors are discussed. In addition, the challenges in the current stage of application are described and future prospects are presented.  This review needs to be revised carefully and hopefully the comments below will be able to help to further improve the review.

  1. An abstract is often presented separately from the article, so it must be able to stand alone. Hence the problem statement, aim, novelty and the recent gap in this field have to be all included.

Response: Thank you for your valuable comment and suggestion. We have a comprehensive summary and outlook on the article, and have conducted a more insightful summary and reflection.

Correction is marked with red color in line 14 to 17, 23 to 25 of the revised manuscript.

  1. Graphical abstract: Not provided, it will be interesting for the author to present the review through figure for better illustration purpose.

Response: Thank you for your valuable comment and suggestion. We have refined the graphical abstract.

  1. Figures need to be supplied at higher resolution.

Response: Thank you for your valuable comment and suggestion. We have modified all the Figures in the article accordingly, and the quality of the Figures and labels in the Figures have been significantly improved.

  1. Introduction: The author could elaborate on the importance of optical biosensor and the advantage of its application for biomolecular detection.

Response: Thank you for your valuable comment and suggestion. We reviewed the importance of optical sensors to understand the advantages of optical biosensors in the field of biomolecular detection and added them to the revised manuscript.

Correction is marked with red color in line 48 to 66 of the revised manuscript.

  1. Introduction: The author could also briefly introduce the different applications of graphene due to its remarkable properties. Kindly refer to the relevant papers:
  2. “A proof of concept: detection of avian influenza H5 gene by a graphene-enhanced electrochemical genosensor”
  3. “Graphene-based field-effect transistor biosensors for the rapid detection and analysis of viruses: A perspective in view of COVID-19”
  4. “Label-free terahertz microfluidic biosensor for sensitive DNA detection using graphene-metasurface hybrid structures”

Response: Thank you for your valuable comment and suggestion. We have reviewed different areas of graphene applications and added them to the revised manuscript.

Correction is marked with red color in line 38 to 43 of the revised manuscript.

  1. Section 2 is too general and too short to be a standalone section, it is suggested that the author could simplify it and combine it with Section 3.

Response: Thank you for your valuable comment and suggestion. We have thoroughly checked and revised the manuscript and some changes have been made to improve the manuscript.

Correction is marked with red color in line 137 to 178 of the revised manuscript.

  1. Section 3: The synthesis approach for graphene and its derivatives has been summarized in many previous reviews, the author could provide a brief summary and discuss on the properties of graphene and its derivatives synthesized using different approaches.

Response: Thank you for your valuable comment and suggestion. We summarized the common methods for preparing graphene and its derivatives and analyzed the advantages and disadvantages to present them in the form of a table.

Correction is marked in line 623 of the revised manuscript.

  1. Subsection titles for Section 4 should be edited accordingly, eg. Section 4.1 should be “graphene and its derivatives-based FRET biosensors”.

Response: Thank you for your valuable comment and suggestion. We have thoroughly checked and revised the manuscript and some changes have been made to improve the manuscript.

Correction is marked with red color in line 703,779,849,923 of the revised manuscript.

  1. Line 587: The scientific names of bacteria (genus and species name) should be italicized.

Response: Thank you for your valuable comment and suggestion. We have thoroughly checked and revised the manuscript and some changes have been made to improve the manuscript.

Correction is marked with red color in line 728,732,771,822,825,878,918 of the revised manuscript.

  1. The biosensing performance of the recent advancement in the graphene and derivatives-based optical biosensor should be tabulated for easy referencing.

Response: Thank you for your valuable comment and suggestion. We have taken the reviewer's suggestion and table for graphene-derived optical biosensors based on graphene have been included in the revised manuscript.

Correction is marked in line 985 of the revised manuscript.

  1. 11. It would be good if the author could provide some take home messages after summarizing the works done on the graphene and derivatives-based optical biosensor. For example, which graphene or the derivative is more suitable for a certain type of optical biosensor and what are the advantage and disadvantage, etc.

Response: Thank you for your valuable comment and suggestion. We have thoroughly checked and revised the manuscript and after summarizing the research results of graphene and its derivatives optical biosensors, some new understandings have emerged.

Correction is marked with red color in line 987 to 1017 of the revised manuscript.

  1. Line 804: The sentence should be changed to “… widely used to modify biosensor/ … widely used in biosensor”.

Response: Thank you for your valuable comment and suggestion. We have thoroughly checked and revised the manuscript and some changes have been made to improve the manuscript.

Correction is marked with red color in line 1021 of the revised manuscript.

  1. The author needs to discuss more in-depth for the possible future prospect in overcoming the challenges of graphene and derivative-based optical biosensor.

Response: Thank you for your valuable comment and suggestion. We have thoroughly checked and revised the manuscript to explore in more depth the challenges that graphene and its derivative optical biosensors may overcome in the future.

Correction is marked with red color in line 1065 to 1093of the revised manuscript.

Round 2

Reviewer 2 Report

Comments have been addressed.
The author needs to properly check the references as some of the reference number are not consistent with the reference cited.